# BLOCK-WISE CODEWORD EMBEDDING FOR RELIABLE MULTI-BIT TEXT WATERMARKING

## ABSTRACT

Recent multi-bit watermarking methods for large language models (LLMs) have focused primarily on maximizing extraction rates. However, our reproduction studies reveal a critical limitation: these approaches suffer from unacceptably high false positive rates (FPR) that undermine their practical deployment. Specifically, existing multi-bit encoding schemes like RS-Watermark achieve high true positive rates even with insertion/deletion attacks but exhibit FPR exceeding 0.90, rendering them unreliable for real-world applications. We propose a robust multi-bit text watermarking framework that addresses this reliability challenge through two key innovations: (i) block-wise error correction that embeds complete codewords within independent text segments, localizing the impact of edits and preventing cascade failures, and (ii) window-shifting detection that systematically recovers codewords despite insertion/deletion-induced misalignments. Our method verifies watermark presence by confirming recovery of the initially embedded codewords, significantly reducing false positives while maintaining high detection accuracy. Experiments on OPT-1.3B and LLaMA-3.2-3B demonstrate substantial improvements over existing multi-bit methods. Under 10% synonym substitution attacks on 200-token texts, our approach achieves TPR of 0.965 with FPR of 0.02 (Precision: 0.9797), compared to RS-Watermark's TPR of 0.97 with FPR of 0.925 (Precision: 0.5132). The framework is code-agnostic, supports progressive detection from partial text, and provides theoretical guarantees for false-positive control. These results establish our method as a practical solution for reliable multi-bit watermarking in production environments.

## 1 INTRODUCTION

Large language models (LLMs) have transformed content generation across creative, professional, and scientific domains, yet raise critical concerns about provenance and potential misuse for deceptive content Solaiman et al. (2019); Bender et al. (2021). Reliably distinguishing human-authored from AI-generated text has become essential for academic integrity, journalism, legal proceedings, and platform governance Mitchell et al. (2023); Gehrmann et al. (2019).

Text watermarking addresses this challenge by embedding imperceptible data into AI-generated content during generation Kirchenbauer et al. (2023). Unlike post-hoc detection methods relying on statistical artifacts Mitchell et al. (2023); Su et al. (2023), watermarking provides stronger origin guarantees while preserving fluency and style.

Watermarking approaches divide into zero-bit (checking watermark presence) and multi-bit (encoding extractable metadata). The green/red partition strategy of Kirchenbauer et al. (2023) biases generation toward a keyed "green" vocabulary subset. Recent multi-bit methods augment partitioning with error-correcting codes (ECCs) to embed message bits. Qu et al. (2025) encodes payloads with Reed-Solomon codes, while Chao et al. (2024) uses LDPC codes with sliding windows for short texts.

Despite strong extraction rates, prior multi-bit schemes exhibit *unacceptably high false positive rates (FPR)*, undermining practical deploymentFu & Russell (2025). Our reproductions, conducted using the official implementation released by Qu et al. (2025), show that under 10% synonym substitution on 200-token texts, Qu et al. (2025) achieve TPR $\approx 0.97$ but FPR $\approx 0.925$ (precision $\approx 0.51$), frequently misclassifying unmarked text as watermarked. In contrast, our method achieves

TPR $= 0.965$ with FPR $= 0.02$ (precision $= 0.9797$). Structurally, the high FPR stems from decoding strategies that treat *any* valid codeword as evidence, irrespective of whether it matches the embedded initially message, and from global synchronization dependencies that collapse under insertions/deletions.

We introduce a robust multi-bit framework that *simultaneously* achieves high TPR and low FPR by (i) embedding *complete codewords in independent blocks* to localize errors and prevent cascade failures under edits, and (ii) deploying a *window-shifting detector* that systematically realigns and recovers codewords after insertion/deletion-induced desynchronization. Crucially, detection verifies that a recovered codeword equals the *designated* codeword that was actually embedded in that block, thereby suppressing spurious matches that inflate FPR. This design achieves both a high TPR and a significantly lower FPR compared to previous multi-bit methods, making them more suitable for real-world forensic applications. The framework is *code-agnostic*: while we instantiate with BCH codes for efficiency and clarity, the design extends to RS/LDPC codes, enabling adaptation to application-specific error patterns.

Our design closes the reliability gap in multi-bit watermarking. On 200-token texts under 10% synonym substitutions, we achieve TPR $= 0.965$ at FPR $= 0.02$, contrasting sharply with prior methods, which have an FPR greater than $0.9$. The incremental detection capability enables progressive verification from partial text, quantifying watermark strength even when some blocks are corrupted, thus broadening real-world deployability.

## 1.1 CONTRIBUTIONS

This work makes the following key contributions:

1. **Low-FPR multi-bit watermarking**: A framework that *significantly* reduces FPR while preserving high TPR, overcoming a critical limitation in recent multi-bit methods and enabling reliable forensic deployment.

2. **Incremental detection framework**: Watermark evidence accumulates from multiple independent codeword segments, enabling graduated confidence assessment rather than binary detection.

3. **Theory for reliability**: Finite-sample bounds and design rules that control false positives and characterize detection power under realistic noise/edit models.

4. **Comprehensive validation**: Experiments across datasets (C4, OpenGen) and model families (OPT-1.3B, LLaMA-3.2-3B) showing state-of-the-art TPR–FPR trade-offs and robustness to substitution/insertion/deletion. For instance, under a 10% synonym substitution attack, a recent method exhibits an FPR of 0.925, whereas our method reduces it to 0.02.

5. **Code-agnostic design**: Compatibility with multiple linear codes (BCH/RS/LDPC), enabling tailoring to domain-specific error patterns.

**Organization.** Section 2 summarizes related work, Section 3 presents the proposed algorithms, Section 4 provides reliability bounds and design guidance, Section 5 presents empirical results, and Section 6 concludes with future research directions.

## 2 RELATED WORKS

Text watermarking for LLMs has rapidly diversified alongside model capabilities and deployment contexts. We organize prior work by *detection objective*: (i) *zero-bit* watermarking, which only tests for the presence of a watermark, and (ii) *multi-bit* watermarking, which embeds and extracts a payload. This lens clarifies robustness requirements (synchronization, error tolerance) and evaluation protocols, and it better reflects recent cryptographic developments, including zero-bit constructions based on pseudorandom error-correcting codes.

### 2.1 ZERO-BIT WATERMARKING

A canonical approach is the keyed green/red partition of Kirchenbauer et al. (2023), which biases generation toward a secret per-token green set and applies a binomial-style hypothesis test at de-

Table 1: Comparison of zero-bit and multi-bit watermarking methods.

| Zero-bit Methods | ECC | Key Features | Limitations |
|---|---|---|---|
| Kirchenbauer et al. (2023) (KGW) | No | Green/red partition; simple test | Fragile to paraphrasing; weak on short texts |
| Wu et al. (2023) (DiPmark) | No | Distribution-preserving; better quality | Reduced watermark strength |
| Zhao et al. (2023) | No | Unigram watermark; provable robustness | Limited to unigram patterns |
| Takezawa et al. (2025) | No | Detectability conditions | No practical robustness |
| Christ & Gunn (2024) | Yes | Pseudorandom ECC; hidden test | Computational overhead; no payload |

| Multi-bit Methods | ECC | Key Features | Limitations |
|---|---|---|---|
| Yoo et al. (2023) | No | Embeds via keywords/syntax | Low extraction accuracy (49.2% at 32-bit) |
| Qu et al. (2025) | Yes (RS) | RS code encoding; high TPR | FPR $\approx$0.9 under insertion/deletion |
| Chao et al. (2024) | Yes (LDPC) | Sliding-window; strong on short text | High FPR risk; complex decoding |

tection. Variants preserve the model distribution to improve quality Wu et al. (2023), or provide robustness under bounded edits Zhao et al. (2023). Exponential reweighting and detectability criteria further sharpen the theory Takezawa et al. (2025). From a cryptographic angle, Christ & Gunn (2024) constructs pseudorandom ECCs whose neighborhoods are indistinguishable from random, enabling hidden presence tests at constant error. Despite their efficiency, most zero-bit schemes rely on aggregate frequency signals and lack explicit synchronization, making them vulnerable to paraphrasing, translation, or token-level desynchronization, especially in short texts.

## 2.2 MULTI-BIT WATERMARKING

Multi-bit watermarking seeks to embed a payload that can be *decoded*. Two broad families appear.

**(a) Non-ECC multi-bit ideas.** Yoo et al. (2023) use invariant features (keywords/syntax) for robustness, but suffer allocation imbalance and low accuracy on longer messages (49.2% match rate for 32-bit).

**(b) ECC-based message encoding.** Qu et al. (2025) pioneered the *ECC-based message-encoding*, which encodes the payload with Reed–Solomon (RS), distributes symbols via pseudorandom segments, and decodes by cracking noisy segment votes to the nearest codeword. Chao et al. (2024) extends this line with LDPC and sliding windows, reporting strong performance on short texts via adaptive biasing and sophisticated decoding.

**Limitations.** ECC-based methods often behave like *message extractors*, not calibrated detectors: nearest-codeword decoding maps even unwatermarked text to valid codewords, driving FPR high—particularly under insertions/deletions or synonym edits. Fu & Russell (2025) formalize this *false detection problem*: conflating detection with identification effectively enlarges key capacity and degrades reliability.

## 2.3 ATTACKS AND EVALUATION PROTOCOLS

Attacks include (i) **substitutions** (synonyms, back-translation, model paraphrasing) Morris et al. (2020); Wieting & Gimpel (2018); Krishna et al. (2023), (ii) **insertions/deletions** that break token–bit alignment, and (iii) **semantic rewrites** that alter surface form while preserving meaning Wolff et al. (2023). While recent frameworks standardize protocols and metrics Pan et al. (2023); Kuditipudi et al. (2023), insertion/deletion scenarios remain underexplored. Prior pseudo-random embedding strategies Yoo et al. (2023); Qu et al. (2025) mitigate—but do not resolve—synchronization, and segment-level voting can yield unacceptably FPR on unwatermarked text.

## 2.4 POSITIONING OF OUR WORK

We address the above gaps with an **incremental detection framework** through: (1) **distributed codeword architecture** embedding *complete* codewords independently, enabling partial recovery and progressive confidence quantification; and (2) **window-shifting detection** that realigns individual codewords, each contributing to accumulated watermark strength. Our *incremental veri-*

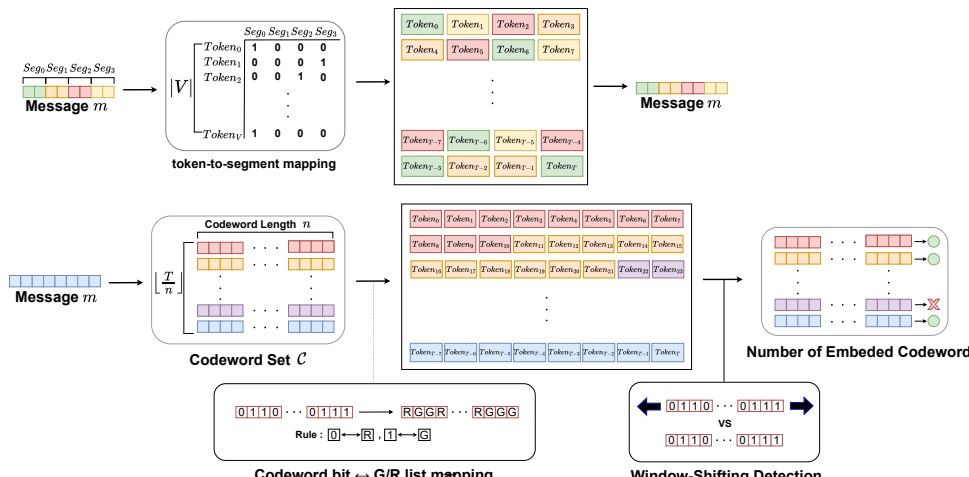

Figure 1: Overview of prior and proposed multi-bit watermarking frameworks. **(Top)** In prior schemes, a deterministic token-to-segment mapping assigns a segment to every token—even for un-watermarked text—so segment votes accumulate and ECC can "correct" noise into a valid codeword, leading to false positives. **(Bottom)** Our incremental detection framework embeds complete code-words in a distributed codeword architecture (realized through token-to-segment and bit–to–G/R list mapping) and employs window-shifting detection with designated verification. This preserves multi-bit payloads while eliminating the "any-codeword" acceptance failure mode, thereby signifi-cantly reducing false positives.

*fication* counts only matching *designated codewords*, transforming binary detection into gradu-ated evidence accumulation while suppressing spurious hits. This quantifies watermark strength continuously—more recovered codewords yield higher confidence. The *code-agnostic* framework (BCH/RS/LDPC/convolutional) achieves substantially improved TPR–FPR through incremental ev-idence collection with explicit insertion/deletion handling. The next section details the algorithms and guarantees.

## 3 Proposed Watermarking Framework

We propose a *reliable multi-bit* watermarking framework that explicitly targets the high FPR pitfall observed in prior multi-bit schemes, while preserving high TPR and robustness to common edits. The method has three pillars: (i) *distributed resilience* via independent codeword blocks that en-able partial watermark recovery and progressive confidence assessment even when some blocks are corrupted, (ii) a *window-shifting* detector that realigns and recovers individual codewords after inser-tion/deletion, contributing to the overall watermark strength score, and (iii) *graduated verification protocol*, which quantifies watermark evidence by counting correctly matched *designated* codewords rather than accepting "any" decodable codeword, thereby enabling continuous watermark strength measurement while suppressing spurious detections.

This incremental approach transforms binary detection into progressive evidence accumulation, where each recovered block contributes to a quantifiable confidence score. This section provides detailed algorithmic descriptions and technical analysis of each component.

### 3.1 Reliable Multi-bit Detection via Designated-Codeword Verification

Prior methods rely on previous tokens to collect information for codeword decoding, treating any text—watermarked or not—identically: the same token contributes to the decoding process, and ECC even corrects "errors" to produce false positives by reconstructing valid codewords from ran-dom noise. In contrast, our approach considers not only tokens but also their relative positions to ver-ify whether patterns match the actual codeword structure, accepting only the designated codeword as

a true detection rather than any valid codeword, thereby significantly reducing false positives while maintaining multi-bit capacity.

### 3.1.1 END-TO-END PROCEDURE

We retain the use of meaningful messages $m \in \{0,1\}^k$ in the watermarking process, but depart from prior approaches by introducing *incremental watermark verification*:

1. **Codeword assignment:** For each block $j$, compute the designated codeword $c^{(j)} \leftarrow \mathbb{E}(m \oplus r^{(j)})$, where $r^{(j)}$ is a key-derived mask used for distance/weight balancing (invertible at detection) as in Algorithm 3.[1]

2. **Distributed embedding.** Embed $c^{(j)}$ in block $j$ via keyed vocabulary partitioning and soft/hard bias as in Algorithm 1, enabling independent recovery of each block.

3. **Shift-aware decoding.** At detection, extract per-block bit strings and perform unique decoding with bounded circular shifts to counter insertion/deletion, treating each recovered block as incremental evidence (Algorithms 4 and 2).

4. **Incremental verification.** Count a block as matched only if the decoded $\hat{c}^{(j)}$ equals the designated $c^{(j)}$; aggregate matches and decide positive if the match ratio exceeds $\theta$.

This preserves the semantics of multi-bit watermarking (the payload can be reconstructed by unmasking $r^{(j)}$ for matched blocks) while *eliminating* the principal FPR failure mode of "any-codeword" acceptance.

Overall, our design offers several advantages: (1) enforces rigorous verification based on codeword matching to maintain low FPR, (2) enables parallel processing of independent blocks, (3) provides graceful degradation under attacks, and (4) supports progressive detection from partial text.

## 3.2 DISTRIBUTED CODEWORD EMBEDDING

### 3.2.1 VOCABULARY PARTITIONING STRATEGY

Following the established approach of Kirchenbauer et al. (2023), we partition the vocabulary $\mathcal{V}$ into two disjoint sets for each block. For block $j$, we compute a block-specific seed such that $\text{seed}_j = H(\mathcal{K}, j)$, where $H$ is a cryptographic hash function (e.g., SHA-3) and $\mathcal{K}$ is the secret watermarking key. Using $\text{seed}_j$, we deterministically partition the vocabulary as $\mathcal{L}_0^{(j)} = \{v \in \mathcal{V} : H(\text{seed}_j, v) \bmod 2 = 0\}$ and $\mathcal{L}_1^{(j)} = \{v \in \mathcal{V} : H(\text{seed}_j, v) \bmod 2 = 1\}$. This block-specific partitioning prevents adversaries from inferring vocabulary assignments across multiple generations, even with partial knowledge of the partitioning strategy.

### 3.2.2 CODEWORD GENERATION AND SELECTION

We pre-compute a set of diverse codewords to avoid statistical patterns that could be exploited by adversaries. Specifically, the codeword generation strategy serves two purposes: (1) excluding all-zero codewords prevents degenerate cases that could impact detection accuracy, and (2) generating codeword pairs with maximum Hamming distance enhances robustness by ensuring diverse bit patterns. The detailed generation procedure is provided in Appendix B.1.

### 3.2.3 DISTRIBUTED EMBEDDING ALGORITHM

Our embedding algorithm processes text generation in blocks of length $n$ tokens, where each block embeds exactly one codeword. Algorithm 1 provides the complete procedure.

**Soft vs. Hard Embedding Schemes**: The soft scheme adds bias $\delta$ to target list logits before applying softmax, allowing natural variation while encouraging codeword-consistent tokens. The hard scheme restricts sampling entirely to the target list, ensuring perfect codeword embedding at a potential cost

---

[1]If an application does not carry a payload, one may set $m = 0^k$ and use only $r^{(j)}$ for per-block variability; the detector remains unchanged. We emphasize our *default* use is multi-bit payloads.

to text quality. The choice between schemes provides a tunable trade-off between watermark strength and naturalness.

| **Algorithm 1** Distributed Watermark Embedding | **Algorithm 2** Window Shifting Detection |
|---|---|
| **Require:** Prompt, key $\mathcal{K}$, codeword queue $\mathcal{Q}$, code length $n$, bias $\delta$, scheme $\in \{$soft, hard$\}$ | **Require:** Text, $\mathcal{Q}$, $s_{\max}$, threshold $\theta$, block length $n$ |
| **Ensure:** Watermarked tokens | **Ensure:** Detection decision |
| 1: $N_p \leftarrow$ prompt length | 1: extract blocks $\mathcal{B}$ (Alg. 4) |
| 2: **for** $t = 0, 1, 2, \ldots$ **do** | 2: matched $\leftarrow 0$, total $\leftarrow \min(\|\mathcal{B}\|, \|\mathcal{Q}\|)$ |
| 3:     logits $\ell^{(t)} \leftarrow$ LM | 3: $\mathcal{S} \leftarrow \{0\} \cup \{-s_{\max}, \ldots, -1, 1, \ldots, s_{\max}\}$ |
| 4:     **if** $t < N_p$ **then** | 4: **for** $i = 0, \ldots,$ total $- 1$ **do** |
| 5:         sample from softmax($\ell^{(t)}$); **continue** | 5:     $b \leftarrow \mathcal{B}[i]$, $c \leftarrow \mathcal{Q}[i]$; *success*$\leftarrow$False |
| 6:     **end if** | 6:     **for** $s \in \mathcal{S}$ **do** |
| 7:     $(j, b) \leftarrow$ divmod$(t - N_p, n)$ | 7:         $b^{(s)} \leftarrow$ ROTATE$(b, s \bmod n)$ |
| 8:     **if** $b = 0$ **and** $\mathcal{Q}[j]$ uninit **then** | 8:         $\hat{c} \leftarrow$ SAFEDECODE$(b^{(s)})$ (Alg. 5) |
| 9:         choose $c \in \mathcal{C}$; $\mathcal{Q}[j] \leftarrow c$ | 9:         **if** $\hat{c} = c$ **then** |
| 10:         seed$_j \leftarrow H(\mathcal{K}, j)$; build $\left(\mathcal{L}_0^{(j)}, \mathcal{L}_1^{(j)}\right)$ | 10:             matched $\leftarrow$ matched $+ 1$; *success*$\leftarrow$True |
| 11:     **end if** | 11:             **break** |
| 12:     $z^{(t)} \leftarrow \mathcal{Q}[j][b]$; $\ell' \leftarrow \ell^{(t)}$ | 12:         **end if** |
| 13:     add $+\delta$ to $\ell'_k$ for $k \in \mathcal{L}_{z^{(t)}}^{(j)}$; | 13:     **end for** |
| 14:     **if** scheme=hard **then** | 14: **end for** |
| 15:         $\ell'_k \leftarrow -\infty$ for $k \notin \mathcal{L}_{z^{(t)}}^{(j)}$ | 15: match_ratio $\leftarrow$ matched/total |
| 16:     **end if** | 16: **return** match_ratio $\geq \theta$ |
| 17:     sample $s^{(t)} \sim$ softmax($\ell'$) | |
| 18: **end for** | |

## 3.3 Window Shifting for Incremental Evidence Recovery

The key innovation enabling *incremental detection* under insertion/deletion attacks is our window shifting mechanism. When tokens are inserted or deleted within a block, the extracted bit sequence becomes a cyclic shift of the original codeword. Our detection algorithm systematically searches for and recovers individual codewords.

**Bit Extraction and Segmentation.** Given the candidate text, we obtain the corresponding bit sequence using the same vocabulary partitioning strategy as in the embedding stage. Each token $s_t$ is mapped to a bit $b_t$ according to its block-specific partition, and the resulting sequence is segmented into blocks of length $n$ as $b_t = f_j(s_t)$, $(j, b) = $ divmod$(t - N_p, n)$, $\mathcal{B} = \{b_{[0:n]}, b_{[n:2n]}, \ldots\}$. Each block represents an independent detection unit for incremental evidence accumulation. The full extraction algorithm, including seed initialization and lookup construction, is provided in Appendix B.2.

**Safe Decoding with Error Handling.** To ensure robust *incremental detection* even when individual blocks contain uncorrectable errors, we implement a safe decoding subroutine that gracefully handles decoder failures. The decoder accepts only codewords within the correction radius $t$ and returns None otherwise, preventing spurious matches while allowing other blocks to contribute to the watermark strength score. The concrete decoding procedure and full algorithm are provided in Appendix B.3.

**Incremental Detection via Window Shifting.** Our core detection algorithm 2 augments standard error-correcting decoding with systematic circular shifting, allowing recovery from misalignments caused by token insertions or deletions. **Circular Shifting Rationale**: When $r$ tokens are inserted at position $p$ within a block, all subsequent bits shift left by $r$ positions. Circular left shifting by $r$ positions can recover the original bit pattern, provided the total corruption (including substitution errors) remains within the error-correction capability of the code.

**Error-Correcting Code Selection and Parameterization.** Our framework is agnostic to the specific choice of error-correcting code. We primarily adopt BCH codes due to their efficiency and well-understood properties, but the framework also accommodates alternatives such as Reed–Solomon, LDPC, or convolutional codes. Detailed selection guidelines and parameterization examples are provided in Appendix C.1.

### 3.4 PARAMETER SELECTION AND OPTIMIZATION

We provide general guidelines for parameter selection, focusing on block length $n$, bias parameter $\delta$, and maximum shift $s_{\max}$. These parameters govern the trade-off between robustness, detection accuracy, and text quality. Comprehensive trade-off analyses and recommended configurations are deferred to Appendix C.2.

### 3.5 COMPUTATIONAL COMPLEXITY ANALYSIS AND SECURITY PROPERTIES

The embedding procedure has the same $O(|\mathcal{V}|)$ per-token complexity as existing methods, while detection introduces an additional factor proportional to the maximum shift $s_{\max}$. Formal derivations and detailed complexity expressions are given in Appendix C.3. Our approach inherits the security guarantees of the underlying hash function and error-correcting code, while introducing additional resilience via block-wise embedding and codeword diversity. A full discussion of key security, codeword diversity, and block independence is provided in Appendix C.4.

## 4 ANALYTICAL BOUNDS FOR FPR/FNR IN ECC-BACKED WATERMARKS

This section develops finite-sample bounds for the proposed watermarking scheme based on block-wise *codeword-presence* detection with window shifting. We quantify false positive (FPR) and false negative (FNR) probabilities under general $q$-ary linear codes, and isolate the role of the embedding bias parameter $\delta$ in the soft-embedding regime. We summarize here the setup and key intuition, while deferring detailed theorems and proofs to Appendix D.

**Setup and Notation.** We consider a $q$-ary linear block code $C \subseteq \Sigma^n$ with unique-decoding radius $t$. Each text block embeds a designated codeword via $\delta$-biased sampling from a green/red partition of the vocabulary. Detection is performed by unique decoding with window shifting to counter misalignments. (Detailed definitions in Appendix D.1.)

**False Positives.** We analyze two types of tests: (i) a naïve "any-codeword" presence test, and (ii) the proposed designated-codeword test with window shifting. Theorems 1 and 2 (Appendix D.2, D.3) quantify single-block and aggregate FPR under these schemes, highlighting exponential suppression in the block length $n$ and the number of blocks $M$.

**False Negatives.** The impact of soft embedding ($\delta$-bias) and adversarial edits is modeled via an effective symbol error probability $p_{\text{tot}}$. Theorem 5 (Appendix D.6) shows that the aggregate FNR decays exponentially in $M$ provided $p_{\text{tot}} < t/n$.

**Design Implications.** The combined FPR/FNR bounds yield a clear design rule: choose parameters $(n, t, s_{\max}, \theta, M, \delta)$ so that $\theta$ balances the two Chernoff exponents, and $\delta$ is large enough to keep the embedding error below $t/n$. See Appendix D.8 for proofs, examples, and entropy-based parameter guidelines.

## 5 RESULTS

### 5.1 EXPERIMENTAL SETUP

**Models and Datasets.** We evaluate on two open-source LLMs, OPT-1.3B and LLaMA-3.2-3B, using datasets from Qu et al. (2025): the C4 corpus (large-scale diverse English text) and the Open-Gen dataset (3,000 two-sentence samples from WikiText-103). Unless noted otherwise, OPT-1.3B and C4 are used as defaults.

**Baselines.** We compare with RS-Watermark(Qu et al. (2025)), which uses Reed–Solomon codes. For fairness, we used the official implementation released by the authors and kept all parameter settings identical. We did not include the LDPC-based scheme of Chao et al. (2024) in our comparisons, as no official implementation has been released; we leave its reproduction and evaluation for future work.

**Parameters and Metrics.** For watermark embedding, we adopt $BCH(n{=}31, k{=}6, t{=}7)$, selecting $n{=}31$ as it provides a balanced TPR–FPR trade-off compared to shorter codewords ($n{=}15$, high TPR but high FPR) and longer ones ($n{=}63$, low FPR but poor TPR), as detailed in Appendix E.7. We evaluate both *soft* and *hard* watermarking but adopt *soft* by default for better text quality (Appendix E.3); in the *soft* setting we vary the insertion strength $\delta \in \{1.5, 2.0, 3.0, 6.0\}$. During detection, we use a window-shift range of $s_{\max} \in \{0, 1, 3, 5\}$, chosen based on the analysis in Appendix C.2.3, to recover alignment under insertion or deletion attacks. Our *incremental detection* protocol quantifies watermark strength by counting recovered codewords: by default, a text is deemed watermarked if at least one originally embedded codeword is recovered, but the system reports the total number of matched codewords as a continuous confidence score. This enables progressive watermark verification where detection confidence increases with each recovered block. A stricter threshold ($\geq 2$ matches) reduces FPR but also lowers TPR (Appendix E.5), demonstrating the flexibility of our graduated detection approach. Between the two detector variants, the Naïve version consistently exhibits high FPR, whereas the structured version achieves FPR $\approx 0.0$ with comparable TPR by leveraging incremental evidence accumulation (Appendix E.1). Hence, we adopt the structured detector in all experiments. We evaluate texts truncated to fixed lengths $T \in \{200, 500\}$ and report standard metrics (TPR, FPR, Precision, F1) alongside the watermark strength score for incremental analysis.

## 5.2 SYNONYM SUBSTITUTION ATTACK

We evaluate robustness under synonym substitution, where words in a watermarked text are replaced with semantically equivalent alternatives. Such variations, covering both paraphrasing and obfuscation, may disrupt token–codeword alignment. Our prior analysis showed that larger watermark insertion strengths $\delta$ reduce the bit error rate (BER), yielding more reliable codeword recovery and higher TPR (Appendix E.2). We categorize attacks into three types depending on their effect on token count: **token-preserving** (no change), **token-decreasing** (shorter replacements, deletion-like), and **token-increasing** (longer replacements, insertion-like). Experiments were conducted at substitution rates of 5% and 10% on the C4 dataset, with complete metrics in Appendix E.8. Supplementary experiments on the OpenGen dataset are also reported in Appendix E.4.

### 5.2.1 TOKEN-PRESERVING SYNONYM SUBSTITUTION

Token-preserving replacements maintain token alignment but alter the statistical distribution of sampled tokens. As shown in Figure 2, both Structured-Ours and RS-Watermark achieve high TPR as $\delta$ increases, but RS-Watermark suffers from FPR values close to 1.0 (e.g., 0.930 at $\delta = 3$). In contrast, Structured-Ours keeps FPR near zero, yielding a far clearer separation between watermarked and unwatermarked texts. We observe the same trends on the OpenGen dataset, as detailed in Appendix E.4.

### 5.2.2 TOKEN-ALTERING SYNONYM SUBSTITUTION (DELETION/INSERTION-LIKE)

When replacements alter token counts, codeword–token alignment is disrupted: fewer tokens shift watermark positions forward (deletion-like), while more tokens shift them backward (insertion-like). As shown in Figure 3, increasing the window-shift parameter $s_{\max}$ consistently improves TPR across both weak ($\delta = 1.5$) and strong ($\delta = 6.0$) watermark insertion strengths. For example, under 10% insertion at $T = 500$ with OPT-1.3B at $\delta = 6.0$, TPR improves from about 0.430 at $s_{\max} = 0$ to 0.960 at $s_{\max} = 5$, showing the pronounced benefit of window-shift detection.

Figure 4 further shows that Structured-Ours maintains low FPR even when TPR is comparable to RS-Watermark For instance, at $T = 500$ with $\delta = 3$, both methods achieve near-perfect TPR (1.0 vs. 0.995), but FPR diverges sharply (0.160 vs. 0.945). Similarly, Figure 5 confirms that across

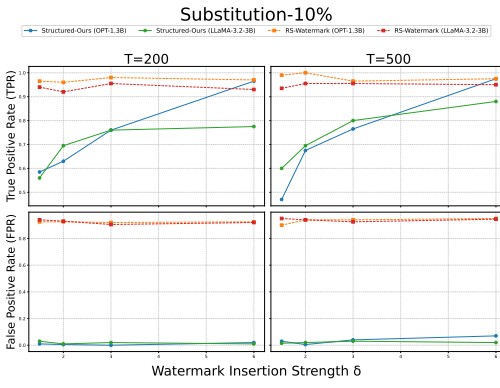

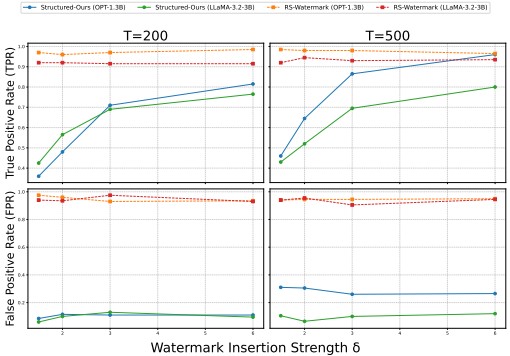

Figure 2: Comparison with RS-Watermark under 10% token-preserving synonym substitution at $s_{\max} = 5$. While maintaining comparable or higher TPR, our method keeps FPR significantly lower, yielding more reliable watermark detection.

Figure 3: Effect of window-shift parameter $s_{\max}$ on TPR under 10% deletion/insertion attacks. Increasing $s_{\max}$ consistently improves TPR for both weak($\delta$=1.5) and strong ($\delta$=6.0) watermarks, demonstrating its effectiveness in mitigating alignment mismatches.

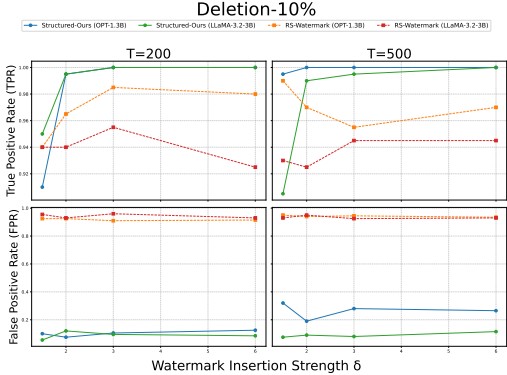

Figure 4: Comparison with RS-Watermark under 10% deletion attacks at $s_{\max} = 5$. Across watermark strengths, our method maintains high TPR while substantially lowering FPR compared to RS-Watermark, indicating more reliable detection.

Figure 5: Comparison with RS-Watermark under 10% insertion attacks at $s_{\max} = 5$. Our method consistently achieves higher TPR and markedly lower FPR than RS-Watermark across different $\delta$ values, demonstrating robustness against insertion-like perturbations.

all insertion strengths, Structured-Ours consistently yields much lower FPR than RS-Watermark, highlighting its superior reliability.

## 5.3 PARAPHRASING ATTACK

As shown in Table 2, Structured-Ours maintains consistently low FPR across all substitution strengths and datasets, even under paraphrasing-based perturbations. Increasing the watermark insertion strength $\delta$ further improves TPR, demonstrating that our method remains robust to semantic rewriting performed by the T5_Paraphrase_Paws model.

In contrast, RS-Watermark exhibits high TPR but suffers from extremely high FPR (often exceeding 90%), making it difficult to reliably distinguish watermarked texts from unwatermarked ones. This highlights a fundamental limitation of existing multi-bit schemes and underscores the necessity of strong false-positive control for practical deployment.

Table 2: Structured-Ours vs. RS-Watermark under paraphrasing (T5_Paraphrase_Paws) on C4 and OpenGen. Structured-Ours maintains low FPR across all $\delta$ and $s_{\max}$ values.

| Setting | | | T200 | | | | | | | |
| --- | --- | --- | --- | --- | --- | --- | --- | --- | --- | --- |
| | | | C4 | | | | OpenGen | | | |
| Model | $\delta$ | $s_{\max}$ | TPR | FPR | Precision | F1_score | TPR | FPR | Precision | F1_score |
| RS-Watermark | 1.5 | - | 0.950 | 0.950 | 0.5000 | 0.6552 | 0.920 | 0.960 | 0.4894 | 0.6389 |
| | 2 | - | 0.920 | 0.930 | 0.4973 | 0.6456 | 0.960 | 0.960 | 0.5000 | 0.6575 |
| | 3 | - | 0.960 | 0.970 | 0.4974 | 0.6553 | 0.960 | 0.920 | 0.5106 | 0.6667 |
| | 6 | - | 0.980 | 0.900 | 0.5213 | 0.6806 | 0.980 | 0.930 | 0.5131 | 0.6735 |
| Structured-Our | 1.5 | 0 | 0.360 | 0.020 | 0.9474 | 0.5217 | 0.450 | 0.020 | 0.9574 | 0.6122 |
| | | 1 | 0.410 | 0.040 | 0.9111 | 0.5655 | 0.400 | 0.000 | 1.0000 | 0.5714 |
| | | 3 | 0.350 | 0.030 | 0.9211 | 0.5072 | 0.570 | 0.050 | 0.9194 | 0.7037 |
| | | 5 | 0.480 | 0.100 | 0.8276 | 0.6076 | 0.480 | 0.100 | 0.8276 | 0.6076 |
| | 2 | 0 | 0.330 | 0.000 | 1.0000 | 0.4962 | 0.520 | 0.010 | 0.9811 | 0.6797 |
| | | 1 | 0.530 | 0.030 | 0.9464 | 0.6795 | 0.540 | 0.020 | 0.9643 | 0.6923 |
| | | 3 | 0.560 | 0.060 | 0.9032 | 0.6914 | 0.710 | 0.030 | 0.9595 | 0.8161 |
| | | 5 | 0.580 | 0.140 | 0.8056 | 0.6744 | 0.530 | 0.130 | 0.8030 | 0.6386 |
| | 3 | 0 | 0.360 | 0.000 | 1.0000 | 0.5294 | 0.600 | 0.000 | 1.0000 | 0.7500 |
| | | 1 | 0.450 | 0.040 | 0.9184 | 0.6040 | 0.630 | 0.060 | 0.9130 | 0.7456 |
| | | 3 | 0.650 | 0.040 | 0.9420 | 0.7692 | 0.710 | 0.060 | 0.9221 | 0.8023 |
| | | 5 | 0.680 | 0.110 | 0.8608 | 0.7598 | 0.720 | 0.040 | 0.9474 | 0.8182 |
| | 6 | 0 | 0.550 | 0.000 | 1.0000 | 0.7097 | 0.550 | 0.010 | 0.9821 | 0.7051 |
| | | 1 | 0.720 | 0.030 | 0.9600 | 0.8229 | 0.780 | 0.080 | 0.9070 | 0.8387 |
| | | 3 | 0.710 | 0.100 | 0.8765 | 0.7845 | 0.700 | 0.070 | 0.9091 | 0.7910 |
| | | 5 | 0.810 | 0.120 | 0.8710 | 0.8394 | 0.910 | 0.070 | 0.9286 | 0.9192 |

## 6 CONCLUSION

In this paper, we proposed an incremental detection framework to overcome the limitations of existing ECC-based watermarking methods. Unlike prior approaches, the proposed scheme effectively suppresses false positives while maintaining stable detection performance under various attacks. Experimental results demonstrate that our method achieves near-zero FPR with consistently high TPR, outperforming the scheme of RS-Watermark and establishing a more reliable watermarking solution. These results highlight the potential of our framework for practical deployment in LLM watermarking. Future research will explore extensions to larger LLMs, more diverse adversarial scenarios, and general linear code structures to further enhance robustness and applicability.

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

## A    LLM Usage Statement

During the preparation of this paper, we used ChatGPT to improve clarity, enhance writing consistency, and assist with grammar refinement. We also used Perplexity.ai to support literature search and discovery (e.g., identifying related work and relevant references). All conceptual contributions, theoretical analyses, model designs, experiments, and research conclusions were developed entirely by the authors.

## B    Detailed Algorithms

### B.1    Diverse Codeword Generation

This algorithm provides the detailed procedure for generating diverse codewords. As described in Section 3, we exclude the all-zero codeword and use maximum-weight pairs to maximize Hamming distance and ensure robustness.

---
**Algorithm 3** Diverse Codeword Generation

---
**Require:** Error-correcting code $\mathcal{C}$ with parameters $(n, k, t)$, secret key $\mathcal{K}$
**Ensure:** Diverse codeword set $\mathcal{Q}$
1: Define message space $\mathcal{M} \leftarrow \{0,1\}^k \setminus \{0^k\}$               ▷ exclude all-zero message
2: $\mathcal{Q} \leftarrow \emptyset$
3: Find maximum weight codeword $c_{\max} = \arg\max_{c \in \mathcal{C}} \text{wt}(c)$
4: **while** $|\mathcal{Q}| <$ required_blocks **do**
5:     Sample random message $m \sim \text{Uniform}(\mathcal{M})$
6:     Encode: $c_1 = \mathcal{E}(m)$
7:     Compute distant pair: $c_2 = c_1 \oplus c_{\max}$
8:     Randomly select $c \in \{c_1, c_2\}$
9:     $\mathcal{Q} \leftarrow \mathcal{Q} \cup \{c\}$
10: **end while**
11: **return** $\mathcal{Q}$

---

### B.2    Bit Sequence Extraction

For completeness, we provide the full pseudocode of the bit extraction procedure that maps generated tokens to binary sequences and segments them into fixed-length blocks. This expands on the conceptual description given in Section 3.

### B.3    Safe Error-Correcting Decoder

This algorithm expands on the safe decoding strategy summarized in Section 3. It ensures robust handling of uncorrectable blocks by returning `None` when decoding exceeds the correction radius.

---

**Algorithm 4** Bit Sequence Extraction

---

**Require:** Text tokens $\{s_0, \ldots, s_T\}$, secret key $\mathcal{K}$, code length $n$
**Ensure:** Blocks $\mathcal{B} = \{b_{[0:n)}, b_{[n:2n)}, \ldots\}$
  1: $N_p \leftarrow$ prompt length;   $U \leftarrow T - N_p + 1$
  2: Initialize bit array $b_{0:U-1}$
  3: $j_{\text{prev}} \leftarrow -1$                                                        $\triangleright$ no block cached yet
  4: **for** $t = N_p, N_p + 1, \ldots, T$ **do**
  5:     $(j, b) \leftarrow \text{divmod}(t - N_p, n)$
  6:     **if** $j \neq j_{\text{prev}}$ **then**                           $\triangleright$ entering a new block: init once
  7:         $\text{seed}_j \leftarrow H(\mathcal{K}, j)$
  8:         Build partitions $\mathcal{L}_0^{(j)}, \mathcal{L}_1^{(j)}$ using $\text{seed}_j$
  9:         Precompute a lookup $f_j(v) \in \{0, 1\}$ for all vocab items $v$
10:         $j_{\text{prev}} \leftarrow j$
11:     **end if**
12:     $b_{t-N_p} \leftarrow f_j(s_t)$                         $\triangleright$ O(1) token-to-bit lookup
13: **end for**
14: **Segmenting:** let $U' \leftarrow \lfloor U/n \rfloor \cdot n$    $\triangleright$ drop incomplete tail; or pad if enabling progressive detection
15: $\mathcal{B} \leftarrow \{\, b_{[0:n)}, b_{[n:2n)}, \ldots, b_{[U'-n:U')} \,\}$
16: **return** $\mathcal{B}$

---

**Algorithm 5** Safe Error-Correcting Decoder

---

**Require:** Bit sequence $x \in \{0, 1\}^n$, code $\mathcal{C}$, correction radius $t$
**Ensure:** $(\hat{c}, d)$ if decodable within $t$; otherwise `None`
  1: **(once per block upstream)** ensure field/parity structures for $\mathcal{C}$ are initialized
  2: **try** $(\hat{c}, d) \leftarrow \text{DECODEWITHDISTANCE}(x)$       $\triangleright$ returns Hamming distance $d$ to $\hat{c}$
  3: **except** any decoder error: **return** `None`
  4: **if** $d \leq t$ **then**
  5:     **return** $(\hat{c}, d)$
  6: **else**
  7:     **return** `None`
  8: **end if**

---

## C ADDITIONAL ANALYSES

### C.1 ERROR-CORRECTING CODE SELECTION AND PARAMETERIZATION

We primarily employ BCH codes Blahut (1983) due to their well-understood properties and efficient implementation. For code length $n = 2^m - 1$, we select parameters based on the trade-off between error-correction capability and false-positive rates:

- **BCH(31,16,3)**: Corrects up to 3 errors, suitable for moderate attack scenarios
- **BCH(63,45,3)**: Longer blocks with same error-correction, better for clean text
- **BCH(127,92,5)**: High error-correction capability for adversarial scenarios

#### C.1.1 ALTERNATIVE ERROR-CORRECTING CODES

Our framework readily accommodates other linear block codes Richardson & Urbanke (2008):

**Reed-Solomon Codes**: Optimal for burst error correction, particularly effective when insertion/deletion attacks create localized corruption patterns.

**LDPC Codes**: Superior performance for longer blocks, but increased computational complexity. Recommended for applications requiring very low false positive rates.

**Convolutional Codes**: Well-suited for streaming applications where text is generated and detected incrementally.

**Code Selection Guidelines**:

- Choose code length $n$ based on expected text length and block granularity requirements
- Select error-correction capability $t$ based on anticipated attack strength
- Balance code rate $k/n$ against false positive requirements using our theoretical analysis (Section 4)

## C.2 PARAMETER SELECTION AND OPTIMIZATION

### C.2.1 BLOCK LENGTH OPTIMIZATION

The choice of block length $n$ involves several trade-offs:

**Shorter blocks** ($n \leq 31$):

- Advantages: Better localization of insertion/deletion effects, faster detection
- Disadvantages: Higher false positive rates, reduced error-correction capability

**Longer blocks** ($n \geq 63$):

- Advantages: Lower false positive rates, stronger error correction
- Disadvantages: Larger vulnerability to insertion/deletion within blocks

We recommend $n = 31$ for most applications, providing a good balance between robustness and efficiency.

### C.2.2 BIAS PARAMETER TUNING

The bias parameter $\delta$ controls the strength of watermark embedding:

- $\delta \in [1.5, 2.0]$: Minimal text quality impact, moderate watermark strength
- $\delta \in [2.0, 2.5]$: Balanced trade-off for most applications
- $\delta \in [2.5, 3.0]$: Strong watermarking for high-security scenarios

### C.2.3 WINDOW SHIFT RANGE

The maximum shift parameter $s_{\max}$ should be chosen based on expected insertion/deletion rates:

$$s_{\max} \geq \alpha \cdot n \cdot p_{\text{ins/del}} \tag{1}$$

where $p_{\text{ins/del}}$ is the expected insertion/deletion rate and $\alpha \geq 1.5$ provides a safety margin. For typical scenarios with $p_{\text{ins/del}} \leq 0.2$, we recommend $s_{\max} = 10$ for $n = 31$.

## C.3 COMPUTATIONAL COMPLEXITY ANALYSIS

### C.3.1 EMBEDDING COMPLEXITY

The computational overhead during text generation consists of:

- Hash computation: $O(1)$ per token
- Vocabulary partitioning: $O(|\mathcal{V}|)$ per block, amortized $O(|\mathcal{V}|/n)$ per token
- Logit modification: $O(|\mathcal{V}|)$ per token

Total embedding complexity: $O(|\mathcal{V}|)$ per token, the same as existing methods.

### C.3.2 DETECTION COMPLEXITY

Detection complexity depends on the number of shift operations:

- Bit extraction: $O(T)$ for text length $T$
- Error correction per block: $O(n^3)$ using standard algorithms
- Window shifting: $O(s_{\max} \cdot n^3)$ per block in worst case

Total detection complexity: $O(T \cdot s_{\max} \cdot n^2)$, where the factor $s_{\max}$ represents the overhead of shift search. For practical parameters ($s_{\max} = 10$, $n = 31$), this remains computationally tractable.

### C.4 SECURITY PROPERTIES

Our method inherits the cryptographic properties of the underlying hash function and error-correcting code while providing additional security benefits through block-wise design.

**Key Security**: The secret key $\mathcal{K}$ determines vocabulary partitioning and codeword selection. Without knowledge of $\mathcal{K}$, an adversary cannot distinguish watermarked from unwatermarked text beyond statistical artifacts due to the one-wayness of the cryptogrpahic hash function.

**Codeword Diversity**: Random codeword generation prevents statistical attacks based on repeated patterns. Each text embeds different codewords, making it infeasible to infer watermarking parameters from multiple samples.

**Block Independence**: Unlike methods that embed single codewords across multiple blocks, our approach ensures that the compromise of one block does not affect others, providing better security compartmentalization.

The following section provides a formal theoretical analysis of detection bounds and false positive rates under our framework.

## D FINITE-SAMPLE BOUNDS: DETAILED PROOFS AND EXAMPLES

This appendix contains the complete derivations, theorems, proofs, and examples for the finite-sample bounds introduced in Section 4.

### D.1 SETUP AND NOTATION

Let $\Sigma = \{0, 1, \ldots, q - 1\}$ and let $C \subseteq \Sigma^n$ be a $q$-ary linear block code with length $n$, dimension $k$, and minimum Hamming distance $d_{\min}$. Its unique-decoding radius is $t = \lfloor (d_{\min} - 1)/2 \rfloor$. Define the $q$-ary Hamming ball volume as

$$V_q(n, t) \triangleq \sum_{i=0}^{t} \binom{n}{i} (q - 1)^i. \tag{2}$$

A text is partitioned into $M$ disjoint blocks. For block $j \in \{1, \ldots, M\}$, a secret seed $\text{seed}_j$ (derived from a global key and block index) deterministically specifies (i) a single *designated* codeword $c^{(j)} \in C$ to be embedded in that block and (ii) a partition of the vocabulary into green/red (or more generally $q$-ary) token lists aligned with the symbols of $c^{(j)}$.

**Embedding.** In *soft* embedding, logits of tokens in the green list are shifted by $+\delta$ while others are left unchanged, and a token is sampled from the resulting softmax. In *hard* embedding, sampling is restricted to the green list (formally, $\delta \to \infty$).

**Detection.** Given a candidate text, the detector extracts a $q$-ary symbol sequence $b^{(j)} \in \Sigma^n$ from each block $j$ according to the green/red partition induced by $\text{seed}_j$, then applies a unique decoder for $C$ to decide whether $b^{(j)}$ lies within Hamming distance $\leq t$ from $c^{(j)}$. To counter local misalignments (e.g., due to in-block insertions/deletions), the detector searches over circular shifts of

magnitude at most $s_{\max}$; denote $S \triangleq 2s_{\max} + 1$ the number of shifts (including zero). The global decision is based on the fraction of blocks that decode successfully: if the match ratio exceeds a threshold $\theta \in (0, 1)$, the text is declared watermarked.

**Stochastic model.** Under $\mathcal{H}_0$ (no watermark), the per-block symbol sequence is modeled as uniformly random in $\Sigma^n$. Under $\mathcal{H}_1$ (watermark present), each block independently suffers symbol errors (from soft embedding and/or adversarial editing) with per-symbol error probability $p_{\text{tot}} \in [0, 1]$, and at most $s_{\max}$ circular misalignment is introduced within the block.

### D.2 NAÏVE "ANY-CODEWORD" PRESENCE TEST

Consider the (undesirable) test that declares a watermark if *there exists* any codeword of $C$ within Hamming distance $t$ of the observed block.

**Theorem 1** (FPR of the any-codeword test). *If $t \leq \lfloor (d_{\min} - 1)/2 \rfloor$ so that Hamming balls of radius $t$ around distinct codewords are disjoint, then under $\mathcal{H}_0$ the single-block false-positive probability of the any-codeword test is*

$$\text{FPR}_{\text{ANY}} = \frac{|C| V_q(n, t)}{q^n} = q^{k-n} V_q(n, t). \tag{3}$$

*Proof.* Under $\mathcal{H}_0$, the block is uniform on $\Sigma^n$. The event "within distance $t$ of *some* codeword" is the disjoint union of the $|C|$ Hamming balls of radius $t$, each of volume $V_q(n, t)$. The probability is therefore $|C| V_q(n, t)/q^n$. $\qquad\square$

**Remark 1** (Binary specialization and magnitude). *For $q = 2$, $V_2(n, t) = \sum_{i=0}^{t} \binom{n}{i}$. Even for modest parameters, the value can be large: e.g., with BCH-like $(n, t) = (31, 3)$ one gets $V_2 = 4{,}992$ and $\text{FPR}_{\text{ANY}} = 2^{k-n} V_2$, which is unacceptably high unless $k \ll n$. For $(n, k, t) = (31, 16, 3)$ BCH codes, $\text{FPR} = 0.152$. This motivates the designated-codeword test below.*

### D.3 DESIGNATED-CODEWORD TEST

Our scheme designates exactly one valid codeword per block $j$ via $\text{seed}_j$; only proximity to this codeword is considered.

**Theorem 2** (Single-block FPR under designated-codeword test). *Under $\mathcal{H}_0$, the single-block FPR for the designated-codeword test equals*

$$p_0 = \frac{V_q(n, t)}{q^n}. \tag{4}$$

*With window shifting over $S$ circular offsets, the FPR obeys the union bound*

$$p_0^{(\text{shift})} \leq \min\{1, S p_0\}. \tag{5}$$

*If the $S$ shifted decoding events are independent (a benign approximation when the decoder's acceptance regions overlap negligibly), then*

$$p_0^{(\text{shift})} = 1 - (1 - p_0)^S = S p_0 + O(p_0^2). \tag{6}$$

*Proof.* For a fixed designated codeword $c^{(j)}$, under $\mathcal{H}_0$ the probability a uniform vector falls within Hamming radius $t$ of $c^{(j)}$ is $V_q(n, t)/q^n$, giving equation 4. Searching $S$ shifts yields at most $S$ chances to fall into a (shifted) acceptance region, whence equation 5. Under independence, the complement probability multiplies across shifts, yielding equation 6. $\qquad\square$

**Remark 2** (Entropy bound). *For any $q$, $V_q(n, t) \leq q^{n H_q(t/n)}$ where $H_q(\cdot)$ is the $q$-ary entropy. Thus*

$$p_0 \leq q^{-n(1 - H_q(t/n))}, \qquad p_0^{(\text{shift})} \lesssim S \, q^{-n(1 - H_q(t/n))}. \tag{7}$$

*This highlights the exponential FPR decay in $n$ at fixed $t/n$.*

**Example 1** (Binary instances). *For $q = 2$ and $(n, t) = (31, 3)$, $p_0 = 3{,}572{,}224/2^{31} \approx 1.6634 \times 10^{-3}$. With $s_{\max} = 10$ ($S = 21$), $p_0^{(\text{shift})} \approx 3.43578 \times 10^{-5}$ via equation 6. For $(n, t) = (63, 3)$ and $(127, 5)$, $p_0 \approx 4.52 \times 10^{-15}$ and $1.56 \times 10^{-30}$, respectively.*

## D.4 AGGREGATE FPR WITH A MATCH-RATIO THRESHOLD

Let $X_j$ be the indicator that block $j$ decodes successfully under $\mathcal{H}_0$. Write $p \triangleq p_0^{(\text{shift})}$.

**Theorem 3** (Aggregate FPR under thresholding). *Assume $\{X_j\}_{j=1}^M$ are independent Bernoulli(p). Then for any $\theta \in (0,1)$,*

$$\Pr_{\mathcal{H}_0}\left[\frac{1}{M}\sum_{j=1}^M X_j \geq \theta\right] \leq \exp\left(-M\,D(\theta\|p)\right), \tag{8}$$

*where $D(a\|b) = a\log\frac{a}{b} + (1-a)\log\frac{1-a}{1-b}$ is the Bernoulli KL divergence.*

*Proof.* This is the standard Chernoff (Cramér–Chernoff) bound for Binomial tails. □

**Remark 3** (Design implication). *Choosing $\theta \gg p$ makes the aggregate FPR exponentially small in $M$. In particular, combining equation 7 and Theorem 3 yields doubly-exponential suppression in $(n, M)$ at fixed $t/n$ and $S$.*

## D.5 SOFT EMBEDDING: SYMBOL ERROR INDUCED BY $\delta$-BIAS

Let $m \in (0,1)$ denote the *pre-bias* total softmax mass of the green list at a generation step. After applying the logit shift $+\delta$ to the green tokens, the probability that the next token is drawn from the green list is

$$P_{\text{green}}(\delta; m) = \frac{me^\delta}{me^\delta + (1-m)} = \sigma\left(\text{logit}(m) + \delta\right), \tag{9}$$

where $\sigma(u) = 1/(1+e^{-u})$ and $\text{logit}(m) = \log(m/(1-m))$.

**Theorem 4** (Per-symbol embedding error in soft mode). *When the designated symbol requires sampling from the green list, the per-symbol embedding error probability is*

$$p_{\text{emb}}(\delta; m) = 1 - P_{green}(\delta; m) = \frac{1-m}{m\,e^\delta + (1-m)}. \tag{10}$$

*In the balanced case $m = \frac{1}{2}$, $p_{\text{emb}}(\delta; \frac{1}{2}) = 1 - \sigma(\delta)$. For a target $p^* \in (0, 1/2)$, it suffices to choose*

$$\delta \geq \log\frac{1-p^*}{p^*} - \text{logit}(m) \tag{11}$$

*to guarantee $p_{\text{emb}}(\delta; m) \leq p^*$.*

*Proof.* It is straightforward from the softmax with a uniform logit shift on the green subset. The inequality is obtained by solving $1 - P_{\text{green}}(\delta; m) \leq p^*$ for $\delta$. □

**Example 2.** *For $m = \frac{1}{2}$, $\delta \in \{2.0, 2.5, 3.0\}$ yields $p_{\text{emb}} \approx \{0.1192, 0.0759, 0.0474\}$, respectively.*

## D.6 DETECTION POWER UNDER EMBEDDING AND ATTACK NOISE

Let $p_{\text{att}} \in [0,1]$ be the adversarial symbol error rate within a block (e.g., substitutions after alignment). A conservative union bound gives $p_{\text{tot}} \leq p_{\text{emb}} + p_{\text{att}}$.

**Theorem 5** (Single-block success and aggregate FNR). *Suppose a block experiences i.i.d. symbol errors with probability $p_{\text{tot}}$ and circular misalignment $\leq s_{\max}$ so that the correct shift is included in the search. Then the single-block success probability is*

$$p_1(n, t, p_{\text{tot}}) = \Pr[\text{Bin}(n, p_{\text{tot}}) \leq t] = \sum_{i=0}^t \binom{n}{i} p_{\text{tot}}^i (1 - p_{\text{tot}})^{n-i}. \tag{12}$$

*If $Y_j$ are i.i.d. indicators of success across blocks under $\mathcal{H}_1$, the aggregate false-negative probability obeys*

$$\Pr_{\mathcal{H}_1}\left[\frac{1}{M}\sum_{j=1}^M Y_j < \theta\right] \leq \exp\left(-M\,D(\theta\|p_1)\right). \tag{13}$$

*Proof.* Unique decoding succeeds iff the number of symbol errors does not exceed $t$; the Binomial tail gives the expression. The Chernoff bound for the lower tail yields the aggregate exponent. □

**Example 3** (Guideline at $(n, t) = (31, 3)$). *With $m = \frac{1}{2}$ and $\delta = 2.5$, $p_{emb} \approx 0.0759$. If $p_{att} \in [0, 0.01]$, then $p_{tot} \in [0.0759, 0.0859]$, giving $p_1 \approx 0.79$ to $0.73$. For $M = 32$ and threshold $\theta = 0.5$, the aggregate FNR is exponentially small by Theorem 5.*

### D.7 SHIFT RECOVERY AND LOCAL EDITS

**Lemma 6** (Sufficient condition for perfect recovery). *If a block suffers at most $s_{max}$ circular shift and at most $t$ symbol substitutions after the correct shift is applied, then window shifting over $S = 2s_{max} + 1$ offsets followed by unique decoding correctly identifies the designated codeword.*

*Proof.* The correct shift lies in the search set, and under that shift the Hamming distance to the designated codeword is $\leq t$. Unique decoding is therefore exact by the definition of $t$. □

**Remark 4** (Modeling in-block insertions/deletions). *When insertions/deletions are confined within a block and do not exceed the shift window, their net effect can be abstracted as a circular shift (alignment) plus residual substitutions. Lemma 6 then applies.*

### D.8 PARAMETER SELECTION VIA ENTROPY BOUNDS

The entropy control in equation 7, together with Theorems 3 and 5, suggests a simple two-sided design: pick $(n, t, s_{max}, \theta, M, \delta)$ so that

$$\underbrace{\exp\left(-M\,D(\theta\|p_0^{(shift)})\right)}_{\text{FPR target }\alpha} \leq \alpha, \qquad p_0^{(shift)} \approx 1 - (1 - p_0)^S, p_0 \leq q^{-n(1 - H_q(t/n))}, \qquad (14)$$

$$\underbrace{\exp\left(-M\,D(\theta\|p_1)\right)}_{\text{FNR target }\beta} \leq \beta, \qquad p_1 = \Pr[\text{Bin}(n, p_{tot}) \leq t], p_{tot} \lesssim p_{emb}(\delta; m) + p_{att}. \quad (15)$$

**Remark 5** (Balanced operation). *A convenient choice is to set $\theta$ near the Chernoff intersection that equalizes exponents $D(\theta\|p_0^{(shift)}) \approx D(\theta\|p_1)$, and to tune $\delta$ to keep $p_{tot} < t/n$ so that $p_1$ remains bounded away from $1/2$.*

### D.9 GENERALIZATIONS

**Proposition 7** (Direct $q$-ary extension). *All the bounds above hold verbatim for $q > 2$ with $V_q(n, t)$ from equation 2; in particular,*

$$p_0 = \frac{V_q(n, t)}{q^n}, \qquad p_0^{(shift)} \leq \min\{1, S\,p_0\}, \qquad \Pr_{\mathcal{H}_0}\left[\text{match ratio} \geq \theta\right] \leq e^{-MD(\theta\|p_0^{(shift)})}.$$

*Proof.* Identical combinatorial counting applies because the unique-decoding radius $t$ depends only on $d_{min}$ and the metric, not on the field size beyond the volume $V_q(n, t)$. □

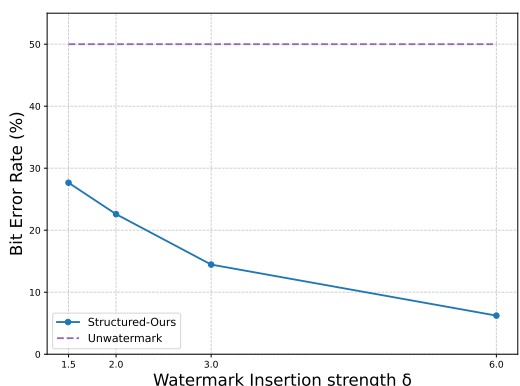

Figure 7: Average Bit Error Rate (BER) as a function of watermark insertion strength $\delta$.

# E SUPPLEMENTARY EXPERIMENTS

## E.1 DETECTION PERFORMANCE WITHOUT ATTACK

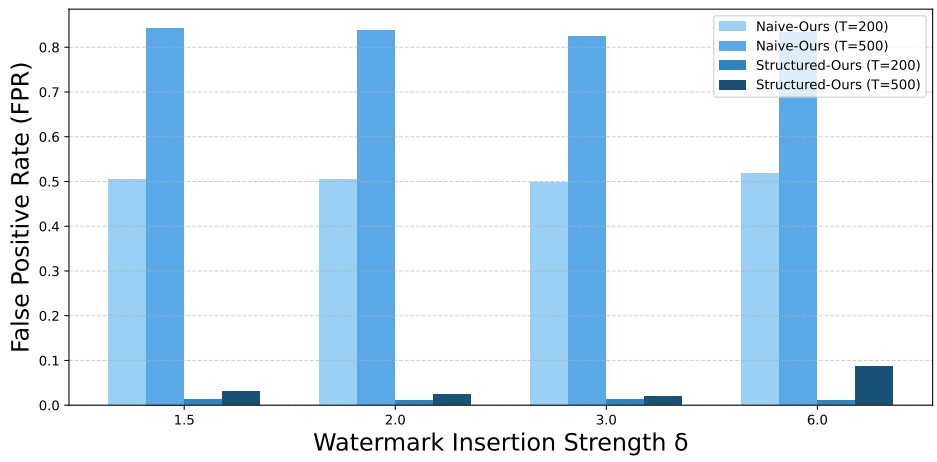

Figure 6: False Positive Rate (FPR) across insertion strengths $\delta$ for Naïve-Ours and Structured-Ours under no-attack settings.

This experiment evaluates detection performance of watermarking techniques in clean environments without adversarial attacks, focusing on how watermark insertion strength $\delta$ and text length $T$ influence reliability.

Figure 6 shows that *Naïve-Ours* suffers from consistently high FPR across all $\delta$ values, failing to distinguish watermarked from unwatermarked text. In contrast, *Structured-Ours* maintains FPR close to zero regardless of $\delta$, demonstrating the effectiveness of structured decoding. For example, at $\delta$=3.0 with $T$=200, *Naïve-Ours* attains TPR=1.000 but FPR=0.499, whereas *Structured-Ours* achieves a comparable TPR=0.987 with FPR=0.013.

## E.2 BIT ERROR RATE ANALYSIS BY WATERMARK INSERTION STRENGTH $\delta$

This experiment evaluates the effect of watermark insertion strength $\delta$ on bit-level codeword reconstruction. As shown in Figure 7, unwatermarked text consistently exhibits about 50% BER, which corresponds to random guessing. In contrast, watermarked text yields significantly lower BERs, with the error rate steadily decreasing as $\delta$ increases. These results demonstrate that stronger watermark

insertion improves the reliability of codeword reconstruction, whereas smaller values of $\delta$ result in BERs closer to random noise, rendering detection more difficult.

## E.3  TEXT QUALITY UNDER WATERMARKING

Table 3: comparison of text quality across watermarking schemes.

| Scheme | $\delta$ | PPL ($\downarrow$) | BLEU ($\uparrow$) | BERTScore(F1) ($\uparrow$) |
|---|---|---|---|---|
| Unwatermark | - | 8.28 | 31.81 | 0.8201 |
| RS-Watermark | 1.5 | 12.78 | 30.99 | 0.8132 |
|  | 2.0 | 13.53 | 22.31 | 0.7740 |
|  | 3.0 | 15.92 | 22.31 | 0.7740 |
|  | 6.0 | 22.16 | 10.14 | 0.6688 |
| Structured-Ours (soft) | 1.5 | 12.41 | 29.14 | 0.8132 |
|  | 2.0 | 13.22 | 27.78 | 0.8082 |
|  | 3.0 | 16.37 | 20.13 | 0.7738 |
|  | 6.0 | 23.67 | 10.16 | 0.6864 |
| Structured-Ours (hard) | - | 28.61 | 6.69 | 0.6312 |

Tables 3 report text quality under a variety of watermarking techniques. Unwatermarked baselines achieve the best overall performance with the lowest performance (PPL), the highest BLEU, and the highest BERTScore, reflecting fluent and semantically faithful outputs.

Qu et al and our structured method (soft) both exhibit the same general trend: As the watermarking insertion strength $\delta$ increases, PPL increases and BLEU and BERTScore decrease, resulting in poor text quality. However, our structured method consistently retains text quality better than Qu et al for all $\delta$ values. For example, when $\delta = 2.0$, our method achieves BLEU score of 27.78 and BERTScore of 0.8082 compared to 22.31 and 0.7740 of Qu et al, respectively. Even in the strongest settings ($\delta = 6.0$, our approach provides slightly higher BLEU and BERTScore.

On the other hand, a hard variant of our method results in severe quality degradation (PPL 28.61, BLEU 6.69 and BERTScore 0.6312), indicating that it is more watermarking than is impractical for quality-sensitive applications. These results highlight that the soft method provides a balanced balance between the robustness of watermarking and text quality.

## E.4  GENERALIZATION STUDY ON THE OPENGEN DATASET

As shown in Figure 8, both methods improve TPR as $\delta$ increases. However, while RS-Watermark consistently exhibits high FPR across all $\delta$ values, our detector keeps FPR close to zero. For example, under a 10% substitution attack, when $T = 500$ and $\delta = 6$, our method achieved an FPR of only 0.015, whereas RS-Watermark reported an FPR of 0.93.

Figure 9 further illustrates that insertion and deletion lead to token–codeword misalignments. Expanding the window size $s_{\max}$ allows the detector to resynchronize with the embedded codewords, thereby significantly improving TPR. In particular, under a 10% insertion attack, when $T = 500$ and $\delta = 6.0$, the TPR improved dramatically from 0.46 at $s_{\max} = 0$ to 0.945 at $s_{\max} = 5$.

Furthermore, as shown in Figures 10 and 11, our method consistently maintains high TPR while keeping FPR low, whereas RS-Watermark achieves high TPR only at the cost of elevated FPR. For example, under a 10% deletion attack, when $T = 500$ and $\delta = 3$, our method achieved perfect detection (TPR = 1.0) with an FPR of only 0.2, while RS-Watermark attained the same TPR (1.0) but suffered from an FPR as high as 0.945.

## E.5  EFFECT OF THRESHOLD INCREASE

This experiment investigates how raising the detection threshold—i.e., the number of required matched codewords—affects detection performance under synonym substitution attacks. Specifi-

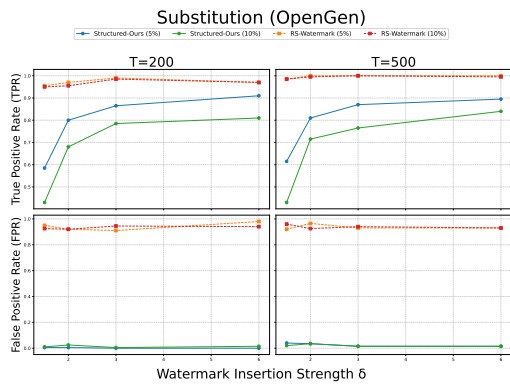

Figure 8: Comparison with RS-Watermark under token-preserving synonym substitution (5% and 10%) at $s_{\max} = 5$. While achieving competitive TPR, our method maintains near-zero FPR across $\delta$, whereas RS-Watermark exhibit consistently high FPR.

Figure 9: Effect of window-shift parameter $s_{\max}$ on TPR under 10% deletion/insertion attacks (OpenGen). Increasing $s_{\max}$ consistently improves TPR for both weak ($\delta = 1.5$) and strong ($\delta = 6.0$) watermarks, demonstrating its effectiveness in recovering alignment mismatches.

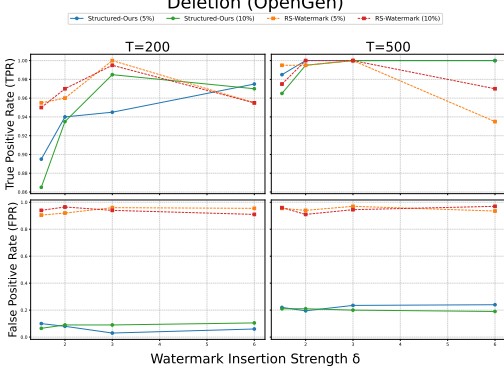

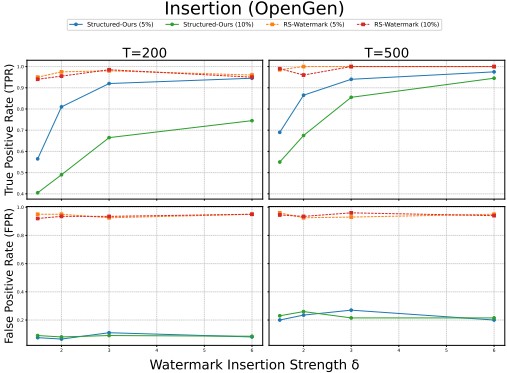

Figure 10: Comparison with RS-Watermark under token-decreasing (deletion-like) substitution (5% and 10%) at $s_{\max} = 5$. Our method sustains high TPR with markedly lower FPR across $\delta$ compared to RS-Watermark

Figure 11: Comparison with RS-Watermark under token-increasing (insertion-like) substitution (5% and 10%) at $s_{\max} = 5$. Our method keeps FPR low while attaining competitive TPR as $\delta$ grows, unlike RS-Watermark whose FPR remains high.

cally, the threshold is increased from requiring at least one matched codeword to requiring at least two.

Figure 12 reports the True Positive Rate (TPR) across different window shift parameters $s_{\max}$ under substitution rates of 5% and 10% at $T = 200$. The results show that stricter thresholds consistently lower TPR across all settings, indicating that some watermarked texts are missed.

Figure 13 presents the corresponding False Positive Rate (FPR). In contrast to TPR, FPR decreases significantly as the threshold increases, with **Structured-Ours-t2** achieving values near zero even under higher substitution rates and large $s_{\max}$. Together, these results highlight the fundamental trade-off: higher thresholds suppress false positives but also reduce TPR.

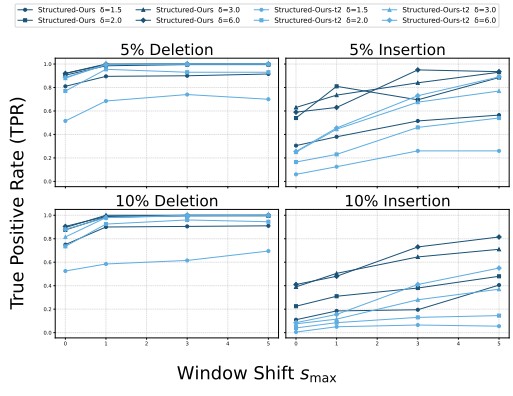 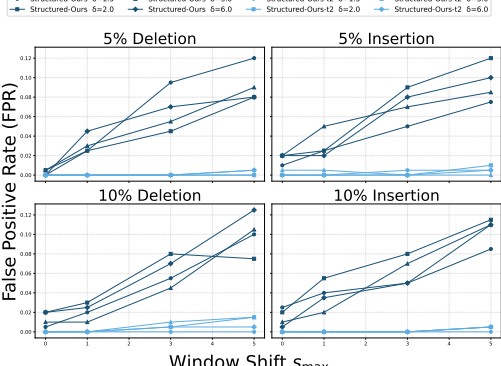

Figure 12: TPR under synonym substitution (rows: 5%, 10%; columns: deletion-like vs. insertion-like, $T = 200$). Increasing the detection threshold from one to two codewords consistently lowers TPR across all $s_{\max}$ and $\delta$.

Figure 13: FPR under synonym substitution (rows: 5%, 10%; columns: deletion-like vs. insertion-like, $T = 200$). Increasing the detection threshold from one to two codewords consistently lowers FPR across all $s_{\max}$ and $\delta$.

### E.6 COMPUTATIONAL COST OF WINDOW-SHIFT DETECTION

The window-shift procedure is applied only during detection, and therefore incurs minimal computational overhead. As shown in Table 4, inference time remains well below one second even with larger codeword lengths $n$ and higher shift budgets $S_{\max}$. This demonstrates that our detection framework is computationally lightweight and practical for real-world use.

Table 4: Inference time (seconds) for varying $S_{\max}$ and codeword lengths $n$.

| Setting | | Inference Time (sec) | | | |
|---|---|---|---|---|---|
| **T** | **n** | $S_{\max} = 0$ | $S_{\max} = 1$ | $S_{\max} = 3$ | $S_{\max} = 5$ |
| | 15 | 0.0252 | 0.0474 | 0.0861 | 0.1262 |
| 200 | 31 | 0.0270 | 0.0526 | 0.1146 | 0.1682 |
| | 63 | 0.0270 | 0.0607 | 0.1345 | 0.2017 |
| | 15 | 0.0394 | 0.0931 | 0.1698 | 0.2741 |
| 500 | 31 | 0.0489 | 0.1127 | 0.2577 | 0.3788 |
| | 63 | 0.0552 | 0.1540 | 0.3431 | 0.5342 |

### E.7 EFFECT OF CODEWORD PARAMETERS ON DETECTION PERFORMANCE

We compared three configurations at $T = 200$: a short codeword ($n = 15, k = 5, t = 3$), a medium codeword ($n = 31, k = 6, t = 7$), and a long codeword ($n = 63, k = 7, t = 15$). The results are summarized in Figure 14.

The short codeword achieved very high TPR but at the cost of large FPR (e.g., under 10% deletion with $\delta = 3.0$, TPR = 1.000 but FPR = 0.945; Table 5). By contrast, the long codeword consistently kept FPR near zero but suffered from reduced TPR, especially under insertion attacks (e.g., under 10% insertion with $\delta = 1.5$, TPR dropped to $0.025$ while FPR remained $0.000$; Table 8).

In between, the medium codeword provided a balanced trade-off, maintaining high TPR while keeping FPR moderate. For this reason, we adopted the medium codeword setting ($n = 31, k = 6, t = 7$) as the default in our main experiments.

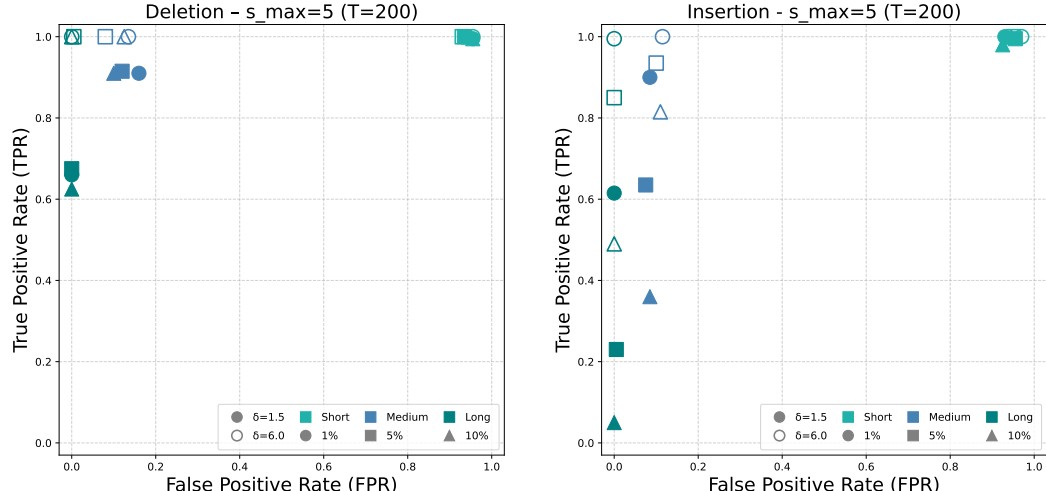

Figure 14: Trade-off between TPR and FPR under token substitution attacks at $T = 200$ ($s_{\max}$=5, $\delta \in \{1.5, 6.0\}$). Left: token-decreasing (Deletion). Right: token-increasing (Insertion). Short codewords ($n$=15) yield higher TPR but increased FPR, long codewords ($n$=63) yield lower FPR but reduced TPR, while medium codewords ($n$=31) provide a balanced trade-off.

Table 5: Detection performance under token-decreasing synonym substitution for BCH codes ($n = 15, k = 5, t = 3$) with $T = 200$.

| Setting | | | 1% Deletion | | | | 5% Deletion | | | | 10% Deletion | | | |
|---|---|---|---|---|---|---|---|---|---|---|---|---|---|---|
| Model | $\delta$ | $s_{\max}$ | TPR | FPR | Precision | F1 | TPR | FPR | Precision | F1 | TPR | FPR | Precision | F1 |
| Structured-Ours (n=15) | 1.5 | 0 | 0.925 | 0.175 | 0.8409 | 0.8809 | 0.880 | 0.230 | 0.7928 | 0.8341 | 0.900 | 0.230 | 0.7965 | 0.8451 |
| | | 1 | 1.000 | 0.530 | 0.6536 | 0.7905 | 0.990 | 0.475 | 0.6758 | 0.8032 | 0.970 | 0.535 | 0.6445 | 0.7745 |
| | | 3 | 0.995 | 0.815 | 0.5497 | 0.7082 | 0.995 | 0.830 | 0.5452 | 0.7044 | 1.000 | 0.800 | 0.5556 | 0.7143 |
| | | 5 | 0.995 | 0.955 | 0.5113 | 0.6746 | 1.000 | 0.935 | 0.5168 | 0.6814 | 0.995 | 0.955 | 0.5103 | 0.6746 |
| | 2.0 | 0 | 0.920 | 0.220 | 0.8070 | 0.8598 | 0.930 | 0.300 | 0.7561 | 0.8341 | 0.895 | 0.180 | 0.8326 | 0.8627 |
| | | 1 | 0.995 | 0.440 | 0.6934 | 0.8173 | 1.000 | 0.500 | 0.6667 | 0.8000 | 0.995 | 0.500 | 0.6656 | 0.7976 |
| | | 3 | 1.000 | 0.815 | 0.5510 | 0.7105 | 1.000 | 0.790 | 0.5587 | 0.7168 | 1.000 | 0.835 | 0.5450 | 0.7055 |
| | | 5 | 1.000 | 0.955 | 0.5115 | 0.6768 | 1.000 | 0.970 | 0.5076 | 0.9734 | 1.000 | 0.930 | 0.5181 | 0.6826 |
| | 3.0 | 0 | 0.915 | 0.200 | 0.8206 | 0.8652 | 0.900 | 0.195 | 0.8219 | 0.8592 | 0.925 | 0.215 | 0.8114 | 0.8645 |
| | | 1 | 1.000 | 0.575 | 0.6350 | 0.7767 | 0.995 | 0.485 | 0.6723 | 0.8024 | 1.000 | 0.525 | 0.6557 | 0.7921 |
| | | 3 | 1.000 | 0.780 | 0.5618 | 0.7194 | 1.000 | 0.860 | 0.5376 | 0.6993 | 1.000 | 0.795 | 0.5571 | 0.7156 |
| | | 5 | 1.000 | 0.945 | 0.5141 | 0.6791 | 1.000 | 0.955 | 0.5115 | 0.6768 | 1.000 | 0.930 | 0.5181 | 0.6826 |
| | 6.0 | 0 | 0.950 | 0.185 | 0.8370 | 0.8899 | 0.945 | 0.215 | 0.8147 | 0.8750 | 0.900 | 0.165 | 0.8451 | 0.8717 |
| | | 1 | 0.995 | 0.510 | 0.6611 | 0.7944 | 1.000 | 0.505 | 0.6645 | 0.7984 | 0.995 | 0.500 | 0.6656 | 0.7976 |
| | | 3 | 1.000 | 0.850 | 0.5405 | 0.7018 | 1.000 | 0.835 | 0.5450 | 0.7055 | 1.000 | 0.830 | 0.5464 | 0.7067 |
| | | 5 | 1.000 | 0.955 | 0.5115 | 0.6768 | 1.000 | 0.930 | 0.5181 | 0.6826 | 1.000 | 0.945 | 0.5141 | 0.6791 |

### E.8 SYNONYM SUBSTITUTION: FULL TABLES AND FIGURES

Figures 15–17 present representative results under 5% synonym substitution attacks, while the main text highlights the 10% case as the most challenging setting. In all figures, results are shown for both OPT-1.3B and LLaMA-3.2-3B, consistently demonstrating that our method achieves comparable or higher TPR and substantially lower FPR than RS-Watermark

Tables 9–20 complement these plots by reporting detailed detection metrics (TPR, FPR, Precision, F1) for each substitution type (token-preserving, deletion-like, insertion-like) across substitution ratios (5%, 10%) and text lengths ($T = 200, 500$), separately for OPT-1.3B and LLaMA-3.2-3B.

### E.9 ROBUSTNESS UNDER MIXED SYNONYM SUBSTITUTION (INSERTION–DELETION–REPLACEMENT)

We further evaluate robustness under a mixed synonym substitution attack, where 20% of the tokens are replaced with synonyms that induce insertion-like, deletion-like, and replacement-like effects

Table 6: Detection performance under token-increasing synonym substitution for BCH codes ($n = 15, k = 5, t = 3$) with $T = 200$.

| Setting | | | 1% Insertion | | | | 5% Insertion | | | | 10% Insertion | | | |
|---|---|---|---|---|---|---|---|---|---|---|---|---|---|---|
| Model | $\delta$ | $s_{max}$ | TPR | FPR | Precision | F1 | TPR | FPR | Precision | F1 | TPR | FPR | Precision | F1 |
| Structured-Ours (n=15) | 1.5 | 0 | 0.875 | 0.245 | 0.7813 | 0.8255 | 0.530 | 0.205 | 0.7211 | 0.6110 | 0.440 | 0.130 | 0.7719 | 0.5605 |
| | | 1 | 0.945 | 0.490 | 0.6585 | 0.7762 | 0.760 | 0.470 | 0.6179 | 0.6816 | 0.625 | 0.500 | 0.5556 | 0.5882 |
| | | 3 | 0.985 | 0.820 | 0.5457 | 0.7023 | 0.940 | 0.815 | 0.5356 | 0.6824 | 0.945 | 0.790 | 0.5447 | 0.6910 |
| | | 5 | 1.000 | 0.930 | 0.5181 | 0.6826 | 0.995 | 0.955 | 0.5103 | 0.6746 | 0.980 | 0.925 | 0.5144 | 0.6747 |
| | 2.0 | 0 | 0.865 | 0.210 | 0.8047 | 0.8337 | 0.690 | 0.195 | 0.7797 | 0.7321 | 0.545 | 0.175 | 0.7569 | 0.6337 |
| | | 1 | 0.980 | 0.505 | 0.6599 | 0.7887 | 0.870 | 0.520 | 0.6259 | 0.7280 | 0.790 | 0.455 | 0.6345 | 0.7038 |
| | | 3 | 1.000 | 0.825 | 0.5479 | 0.7080 | 0.955 | 0.820 | 0.5380 | 0.6883 | 0.940 | 0.775 | 0.5481 | 0.6924 |
| | | 5 | 1.000 | 0.965 | 0.5089 | 0.6745 | 0.995 | 0.945 | 0.5129 | 0.6769 | 0.985 | 0.925 | 0.5157 | 0.6770 |
| | 3.0 | 0 | 0.885 | 0.215 | 0.8045 | 0.8429 | 0.745 | 0.225 | 0.7680 | 0.7563 | 0.620 | 0.230 | 0.7294 | 0.6703 |
| | | 1 | 0.950 | 0.475 | 0.6667 | 0.7835 | 0.890 | 0.480 | 0.6496 | 0.7511 | 0.775 | 0.495 | 0.6102 | 0.6828 |
| | | 3 | 1.000 | 0.815 | 0.5510 | 0.7105 | 0.970 | 0.815 | 0.5434 | 0.6966 | 0.975 | 0.845 | 0.5357 | 0.6915 |
| | | 5 | 1.000 | 0.920 | 0.5208 | 0.6849 | 0.990 | 0.945 | 0.5116 | 0.6746 | 1.000 | 0.930 | 0.5181 | 0.6826 |
| | 6.0 | 0 | 0.905 | 0.195 | 0.8227 | 0.8619 | 0.725 | 0.230 | 0.7592 | 0.7417 | 0.645 | 0.190 | 0.7725 | 0.7030 |
| | | 1 | 0.990 | 0.560 | 0.6387 | 0.7765 | 0.895 | 0.505 | 0.6393 | 0.7458 | 0.860 | 0.505 | 0.6300 | 0.7273 |
| | | 3 | 1.000 | 0.820 | 0.5495 | 0.7092 | 0.990 | 0.825 | 0.5455 | 0.7034 | 0.970 | 0.800 | 0.5480 | 0.7004 |
| | | 5 | 1.000 | 0.970 | 0.5076 | 0.6734 | 1.000 | 0.950 | 0.5128 | 0.6780 | 1.000 | 0.945 | 0.5141 | 0.6791 |

Table 7: Detection performance under token-decreasing synonym substitution for BCH codes ($n = 63, k = 7, t = 15$) with $T = 200$.

| Setting | | | 1% Deletion | | | | 5% Deletion | | | | 10% Deletion | | | |
|---|---|---|---|---|---|---|---|---|---|---|---|---|---|---|
| Model | $\delta$ | $s_{max}$ | TPR | FPR | Precision | F1 | TPR | FPR | Precision | F1 | TPR | FPR | Precision | F1 |
| Structured-Ours (n=63) | 1.5 | 0 | 0.585 | 0 | 1.0000 | 0.7382 | 0.490 | 0 | 1.0000 | 0.6577 | 0.560 | 0 | 1.0000 | 0.7179 |
| | | 1 | 0.715 | 0 | 1.0000 | 0.8338 | 0.650 | 0 | 1.0000 | 0.7879 | 0.640 | 0.005 | 0.9922 | 0.7781 |
| | | 3 | 0.725 | 0 | 1.0000 | 0.8406 | 0.660 | 0 | 1.0000 | 0.7952 | 0.650 | 0 | 1.0000 | 0.7879 |
| | | 5 | 0.660 | 0 | 1.0000 | 0.7952 | 0.675 | 0 | 1.0000 | 0.8060 | 0.625 | 0 | 1.0000 | 0.7692 |
| | 2.0 | 0 | 0.820 | 0 | 1.0000 | 0.9011 | 0.775 | 0 | 1.0000 | 0.8732 | 0.770 | 0 | 1.0000 | 0.8701 |
| | | 1 | 0.955 | 0 | 1.0000 | 0.9770 | 0.920 | 0 | 1.0000 | 0.9583 | 0.935 | 0 | 1.0000 | 0.9664 |
| | | 3 | 0.955 | 0 | 1.0000 | 0.9770 | 0.960 | 0 | 1.0000 | 0.9796 | 0.920 | 0 | 1.0000 | 0.9583 |
| | | 5 | 0.965 | 0.010 | 0.9897 | 0.9772 | 0.960 | 0 | 1.0000 | 0.9796 | 0.940 | 0 | 1.0000 | 0.9691 |
| | 3.0 | 0 | 0.910 | 0 | 1.0000 | 0.9529 | 0.880 | 0 | 1.0000 | 0.9362 | 0.830 | 0 | 1.0000 | 0.9071 |
| | | 1 | 1.000 | 0 | 1.0000 | 1.0000 | 0.990 | 0 | 1.0000 | 0.9950 | 0.975 | 0 | 1.0000 | 0.9873 |
| | | 3 | 1.000 | 0.005 | 0.9950 | 0.9975 | 0.995 | 0 | 1.0000 | 0.9975 | 0.995 | 0 | 1.0000 | 0.9975 |
| | | 5 | 1.000 | 0 | 1.0000 | 1.0000 | 1.000 | 0 | 1.0000 | 1.0000 | 1.000 | 0 | 1.0000 | 1.0000 |
| | 6.0 | 0 | 0.925 | 0 | 1.0000 | 0.9610 | 0.890 | 0 | 1.0000 | 0.9418 | 0.855 | 0 | 1.0000 | 0.9218 |
| | | 1 | 0.995 | 0 | 1.0000 | 0.9975 | 0.980 | 0 | 1.0000 | 0.9899 | 0.995 | 0 | 1.0000 | 0.9975 |
| | | 3 | 1.000 | 0 | 1.0000 | 1.0000 | 1.000 | 0 | 1.0000 | 1.0000 | 1.000 | 0 | 1.0000 | 1.0000 |
| | | 5 | 1.000 | 0 | 1.0000 | 1.0000 | 1.000 | 0.005 | 0.9950 | 0.9975 | 1.000 | 0 | 1.0000 | 1.0000 |

Table 8: Detection performance under token-increasing synonym substitution for BCH codes ($n = 63, k = 7, t = 15$) with $T = 200$.

| Setting | | | 1% Insertion | | | | 5% Insertion | | | | 10% Insertion | | | |
|---|---|---|---|---|---|---|---|---|---|---|---|---|---|---|
| Model | $\delta$ | $s_{max}$ | TPR | FPR | Precision | F1 | TPR | FPR | Precision | F1 | TPR | FPR | Precision | F1 |
| Structured-Ours (n=63) | 1.5 | 0 | 0.440 | 0 | 1.0000 | 0.6111 | 0.110 | 0 | 1.0000 | 0.1982 | 0.025 | 0 | 1.0000 | 0.0488 |
| | | 1 | 0.490 | 0 | 1.0000 | 0.6577 | 0.155 | 0 | 1.0000 | 0.2684 | 0.010 | 0 | 1.0000 | 0.0198 |
| | | 3 | 0.595 | 0 | 1.0000 | 0.7461 | 0.210 | 0 | 1.0000 | 0.3471 | 0.060 | 0 | 1.0000 | 0.1132 |
| | | 5 | 0.615 | 0 | 1.0000 | 0.7616 | 0.230 | 0.005 | 0.9787 | 0.3725 | 0.050 | 0 | 1.0000 | 0.0952 |
| | 2.0 | 0 | 0.635 | 0 | 1.0000 | 0.7768 | 0.240 | 0 | 1.0000 | 0.3871 | 0.065 | 0 | 1.0000 | 0.1121 |
| | | 1 | 0.805 | 0 | 1.0000 | 0.8920 | 0.255 | 0 | 1.0000 | 0.4064 | 0.080 | 0 | 1.0000 | 0.1481 |
| | | 3 | 0.875 | 0 | 1.0000 | 0.9333 | 0.350 | 0 | 1.0000 | 0.5185 | 0.115 | 0 | 1.0000 | 0.2063 |
| | | 5 | 0.915 | 0 | 1.0000 | 0.9556 | 0.485 | 0 | 1.0000 | 0.6532 | 0.145 | 0 | 1.0000 | 0.2533 |
| | 3.0 | 0 | 0.800 | 0 | 1.0000 | 0.8889 | 0.320 | 0 | 1.0000 | 0.4848 | 0.105 | 0 | 1.0000 | 0.1900 |
| | | 1 | 0.900 | 0 | 1.0000 | 0.9474 | 0.515 | 0 | 1.0000 | 0.6799 | 0.145 | 0 | 1.0000 | 0.2533 |
| | | 3 | 0.960 | 0 | 1.0000 | 0.9796 | 0.645 | 0 | 1.0000 | 0.7842 | 0.280 | 0 | 1.0000 | 0.4375 |
| | | 5 | 0.985 | 0 | 1.0000 | 0.9924 | 0.710 | 0 | 1.0000 | 0.8304 | 0.305 | 0.005 | 0.9861 | 0.5221 |
| | 6.0 | 0 | 0.840 | 0 | 1.0000 | 0.9130 | 0.395 | 0 | 1.0000 | 0.5663 | 0.145 | 0 | 1.0000 | 0.2533 |
| | | 1 | 0.900 | 0 | 1.0000 | 0.9474 | 0.530 | 0 | 1.0000 | 0.6928 | 0.250 | 0 | 1.0000 | 0.4000 |
| | | 3 | 0.985 | 0 | 1.0000 | 0.9924 | 0.735 | 0 | 1.0000 | 0.8473 | 0.430 | 0 | 1.0000 | 0.6014 |
| | | 5 | 0.995 | 0 | 1.0000 | 0.9975 | 0.850 | 0 | 1.0000 | 0.9189 | 0.490 | 0 | 1.0000 | 0.6577 |

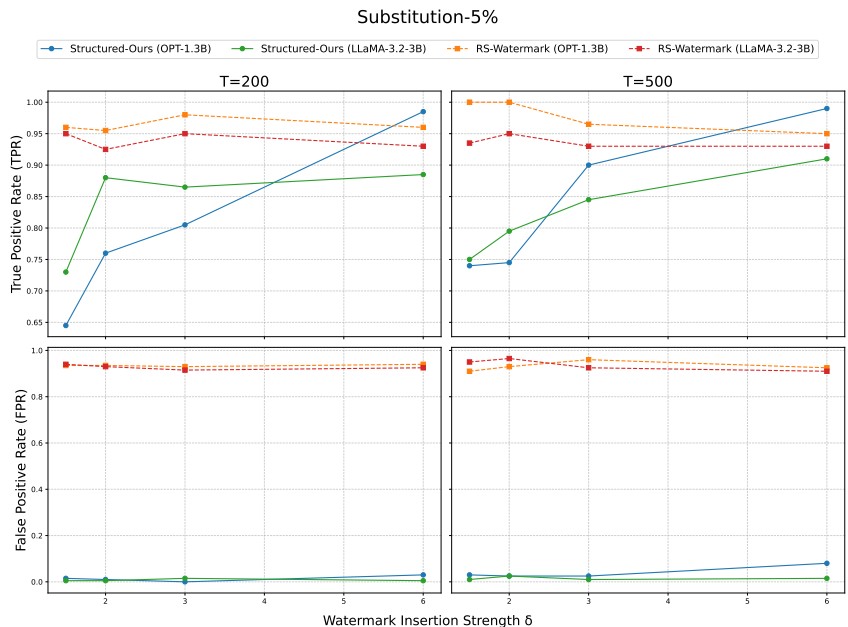

Figure 15: Comparison with RS-Watermark under 5% token-preserving synonym substitution at $s_{\max} = 5$. Both methods achieve high TPR, but RS-Watermark exhibits substantially higher FPR, whereas our method keeps FPR near zero, indicating more reliable watermark detection.

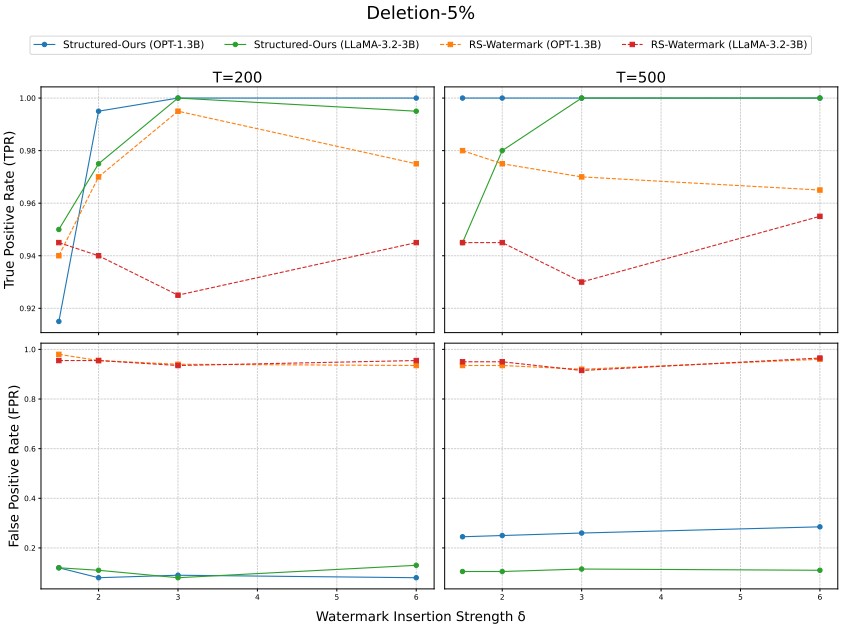

Figure 16: Comparison with RS-Watermark under 5% deletion-like synonym substitution at $s_{\max} = 5$. Our method achieves higher TPR and substantially lower FPR than RS-Watermark, demonstrating that our watermark detector operates more reliably in this challenging setting.

simultaneously. This setting combines the three previously analyzed cases (token-preserving, token-decreasing, and token-increasing) and represents the most challenging form of synonym-based perturbation.

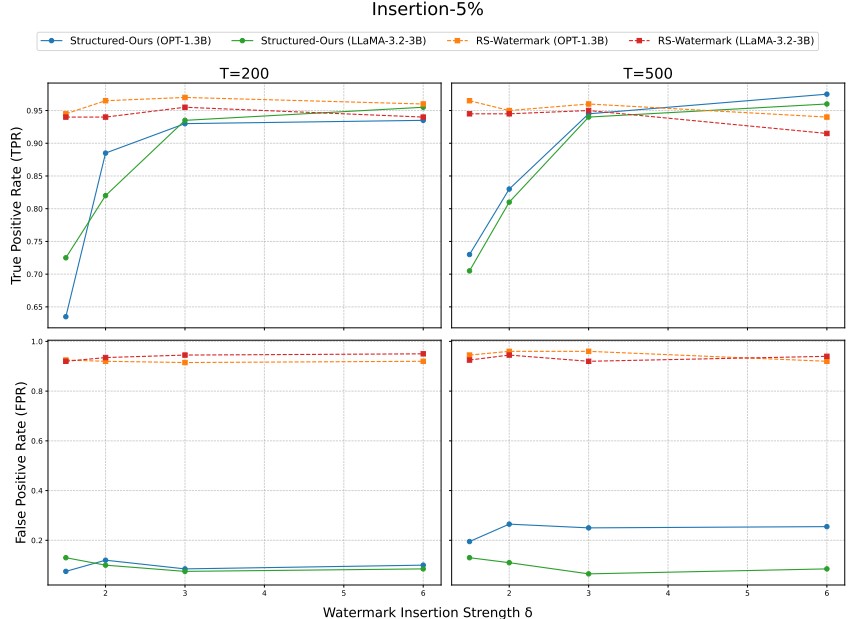

Figure 17: Comparison with RS-Watermark under 5% insertion-like synonym substitution at $s_{\max} = 5$. Our method achieves higher TPR and substantially lower FPR than RS-Watermark, demonstrating that our watermark detector operates more reliably in this challenging setting.

Table 9: Detection performance under 5% token-preserving synonym substitution with OPT-1.3B.

| Setting | | **T200** | | | | **T500** | | | |
|---|---|---|---|---|---|---|---|---|---|
| **Model** | $\delta$ | TPR | FPR | Precision | F1 | TPR | FPR | Precision | F1 |
| | 1.5 | 0.960 | 0.935 | 0.5066 | 0.6632 | 1.000 | 0.910 | 0.5236 | 0.6873 |
| RS-Watermark | 2 | 0.955 | 0.935 | 0.5053 | 0.6609 | 1.000 | 0.930 | 0.5181 | 0.6826 |
| | 3 | 0.980 | 0.930 | 0.5131 | 0.6735 | 0.965 | 0.960 | 0.5013 | 0.6598 |
| | 6 | 0.960 | 0.940 | 0.5053 | 0.6621 | 0.950 | 0.925 | 0.5067 | 0.6609 |
| | 1.5 | 0.645 | 0.015 | 0.9773 | 0.7771 | 0.740 | 0.030 | 0.9610 | 0.8362 |
| Structured-Ours | 2 | 0.760 | 0.010 | 0.9870 | 0.8588 | 0.745 | 0.025 | 0.9675 | 0.8418 |
| | 3 | 0.805 | 0.000 | 1.0000 | 0.8920 | 0.900 | 0.025 | 0.9730 | 0.9351 |
| | 6 | 0.985 | 0.030 | 0.9704 | 0.9777 | 0.990 | 0.080 | 0.9252 | 0.9565 |

Tables 21 and 22 show that Structured-Ours yields lower TPR than RS-Watermark under this strong mixed attack, which is expected given the severe distortions introduced. However, RS-Watermark's apparently stable TPR is misleading: its FPR remains extremely high (often near 1.0), causing the detector to label most texts—including unwatermarked ones—as watermarked. In contrast, Structured-Ours consistently maintains low FPR across all $\delta$ and $s_{\max}$ values for both $T=200$ and $T=500$.

This reliable false-positive control enables clear separation between watermarked and unwatermarked texts even under 20% mixed synonym substitution, highlighting the practical robustness and reliability of our detection framework.

Table 10: Detection performance under 5% token-preserving synonym substitution using LLaMA-3.2-3B.

| Setting | | T200 | | | | T500 | | | |
|---|---|---|---|---|---|---|---|---|---|
| Model | $\delta$ | TPR | FPR | Precision | F1-score | TPR | FPR | Precision | F1-score |
| RS-Watermark | 1.5 | 0.950 | 0.940 | 0.5026 | 0.6574 | 0.935 | 0.950 | 0.4960 | 0.6480 |
| | 2 | 0.925 | 0.930 | 0.4986 | 0.6479 | 0.950 | 0.965 | 0.4961 | 0.6518 |
| | 3 | 0.950 | 0.915 | 0.5093 | 0.6631 | 0.930 | 0.925 | 0.5013 | 0.6514 |
| | 6 | 0.930 | 0.925 | 0.5013 | 0.6514 | 0.930 | 0.910 | 0.5054 | 0.6549 |
| Structured-Ours | 1.5 | 0.730 | 0.005 | 0.9931 | 0.8414 | 0.750 | 0.010 | 0.9868 | 0.8522 |
| | 2 | 0.880 | 0.005 | 0.9943 | 0.9336 | 0.795 | 0.025 | 0.9695 | 0.8736 |
| | 3 | 0.865 | 0.015 | 0.9829 | 0.9202 | 0.845 | 0.010 | 0.9883 | 0.9111 |
| | 6 | 0.885 | 0.005 | 0.9943 | 0.9365 | 0.910 | 0.015 | 0.9837 | 0.9454 |

Table 11: Detection performance under 10% token-preserving synonym substitution with OPT-1.3B.

| Setting | | T200 | | | | T500 | | | |
|---|---|---|---|---|---|---|---|---|---|
| Model | $\delta$ | TPR | FPR | Precision | F1 | TPR | FPR | Precision | F1 |
| RS-Watermark | 1.5 | 0.965 | 0.925 | 0.5106 | 0.6678 | 0.990 | 0.900 | 0.5238 | 0.6851 |
| | 2 | 0.960 | 0.925 | 0.5093 | 0.6655 | 1.000 | 0.940 | 0.5155 | 0.6803 |
| | 3 | 0.980 | 0.920 | 0.5158 | 0.6759 | 0.965 | 0.940 | 0.5066 | 0.6644 |
| | 6 | 0.970 | 0.925 | 0.5132 | 0.6724 | 0.975 | 0.950 | 0.5065 | 0.6667 |
| Structured-Ours | 1.5 | 0.585 | 0.010 | 0.9832 | 0.7335 | 0.470 | 0.030 | 0.9400 | 0.6267 |
| | 2 | 0.630 | 0.005 | 0.9921 | 0.7706 | 0.675 | 0.005 | 0.9926 | 0.8036 |
| | 3 | 0.760 | 0.000 | 1.0000 | 0.8636 | 0.765 | 0.040 | 0.9503 | 0.8476 |
| | 6 | 0.965 | 0.020 | 0.9797 | 0.9723 | 0.975 | 0.070 | 0.9330 | 0.9535 |

Table 12: Detection performance under 10% token-preserving synonym substitution using LLaMA-3.2-3B.

| Setting | | T200 | | | | T500 | | | |
|---|---|---|---|---|---|---|---|---|---|
| Model | $\delta$ | TPR | FPR | Precision | F1-score | TPR | FPR | Precision | F1-score |
| RS-Watermark | 1.5 | 0.940 | 0.940 | 0.5000 | 0.6527 | 0.935 | 0.950 | 0.4960 | 0.6481 |
| | 2 | 0.920 | 0.930 | 0.4979 | 0.6456 | 0.955 | 0.940 | 0.5039 | 0.6597 |
| | 3 | 0.955 | 0.905 | 0.5134 | 0.6678 | 0.955 | 0.925 | 0.5079 | 0.6631 |
| | 6 | 0.930 | 0.920 | 0.5027 | 0.6526 | 0.950 | 0.945 | 0.5013 | 0.6563 |
| Structured-Ours | 1.5 | 0.560 | 0.030 | 0.9491 | 0.7044 | 0.600 | 0.015 | 0.9756 | 0.7430 |
| | 2 | 0.695 | 0.010 | 0.9858 | 0.8152 | 0.695 | 0.020 | 0.9720 | 0.8104 |
| | 3 | 0.760 | 0.020 | 0.9743 | 0.8539 | 0.800 | 0.030 | 0.9638 | 0.8743 |
| | 6 | 0.775 | 0.010 | 0.9869 | 0.8555 | 0.880 | 0.020 | 0.9777 | 0.9263 |

Table 13: Detection performance under 5% token-decreasing synonym substitution with OPT-1.3B.

| Setting | | | T200 | | | | T500 | | | |
|---|---|---|---|---|---|---|---|---|---|---|
| Model | $\delta$ | $s_{\max}$ | TPR | FPR | Precision | F1 | TPR | FPR | Precision | F1 |
| RS-Watermark | 1.5 | - | 0.940 | 0.980 | 0.4896 | 0.6438 | 0.980 | 0.935 | 0.5117 | 0.6724 |
| | 2 | - | 0.970 | 0.955 | 0.5039 | 0.6632 | 0.975 | 0.935 | 0.5105 | 0.6701 |
| | 3 | - | 0.995 | 0.940 | 0.5142 | 0.6780 | 0.970 | 0.920 | 0.5132 | 0.6713 |
| | 6 | - | 0.975 | 0.935 | 0.5105 | 0.6701 | 0.965 | 0.960 | 0.5013 | 0.6598 |
| Structured-Ours | 1.5 | 0 | 0.810 | 0.000 | 1.0000 | 0.8950 | 0.900 | 0.030 | 0.9677 | 0.9326 |
| | | 1 | 0.895 | 0.025 | 0.9728 | 0.9323 | 0.985 | 0.075 | 0.9292 | 0.9563 |
| | | 3 | 0.900 | 0.095 | 0.9045 | 0.9023 | 0.995 | 0.140 | 0.8767 | 0.9321 |
| | | 5 | 0.915 | 0.120 | 0.8841 | 0.8993 | 1.000 | 0.245 | 0.8032 | 0.8909 |
| | 2.0 | 0 | 0.905 | 0.005 | 0.9945 | 0.9476 | 0.905 | 0.015 | 0.9837 | 0.9427 |
| | | 1 | 0.985 | 0.025 | 0.9801 | 0.9801 | 0.995 | 0.060 | 0.9431 | 0.9684 |
| | | 3 | 0.995 | 0.045 | 0.9567 | 0.9755 | 1.000 | 0.130 | 0.8850 | 0.9390 |
| | | 5 | 0.995 | 0.080 | 0.9256 | 0.9590 | 1.000 | 0.250 | 0.8000 | 0.8888 |
| | 3.0 | 0 | 0.920 | 0.005 | 0.9946 | 0.9558 | 0.905 | 0.020 | 0.9784 | 0.9403 |
| | | 1 | 0.995 | 0.030 | 0.9707 | 0.9827 | 0.995 | 0.075 | 0.9299 | 0.9614 |
| | | 3 | 1.000 | 0.055 | 0.9479 | 0.9732 | 1.000 | 0.185 | 0.8439 | 0.9153 |
| | | 5 | 1.000 | 0.090 | 0.9174 | 0.9569 | 1.000 | 0.260 | 0.7937 | 0.8850 |
| | 6.0 | 0 | 0.920 | 0.000 | 1.0000 | 0.9583 | 0.950 | 0.030 | 0.9694 | 0.9596 |
| | | 1 | 1.000 | 0.045 | 0.9569 | 0.9780 | 0.995 | 0.075 | 0.9299 | 0.9614 |
| | | 3 | 1.000 | 0.070 | 0.9346 | 0.9662 | 1.000 | 0.200 | 0.8333 | 0.9091 |
| | | 5 | 1.000 | 0.080 | 0.9259 | 0.9615 | 1.000 | 0.285 | 0.7782 | 0.8753 |

Table 14: Detection performance under 5% token-deleting synonym substitution using LLaMA-3.2-3B.

| Setting | | | T200 | | | | T500 | | | |
|---|---|---|---|---|---|---|---|---|---|---|
| Model | $\delta$ | $s_{\max}$ | TPR | FPR | Precision | F1-score | TPR | FPR | Precision | F1-score |
| RS-Watermark | 1.5 | - | 0.945 | 0.955 | 0.4973 | 0.6517 | 0.945 | 0.950 | 0.4986 | 0.6528 |
| | 2.0 | - | 0.940 | 0.935 | 0.5013 | 0.6539 | 0.945 | 0.955 | 0.4973 | 0.6517 |
| | 3.0 | - | 0.925 | 0.935 | 0.4973 | 0.6468 | 0.930 | 0.915 | 0.5041 | 0.6537 |
| | 6.0 | - | 0.945 | 0.955 | 0.4973 | 0.6517 | 0.955 | 0.965 | 0.4973 | 0.6541 |
| Structured-Ours | 1.5 | 0 | 0.935 | 0.040 | 0.9589 | 0.9468 | 0.820 | 0.015 | 0.9820 | 0.8937 |
| | | 1 | 0.930 | 0.025 | 0.9738 | 0.9514 | 0.915 | 0.010 | 0.9891 | 0.9506 |
| | | 3 | 0.935 | 0.080 | 0.9211 | 0.9280 | 0.935 | 0.045 | 0.9541 | 0.9444 |
| | | 5 | 0.950 | 0.120 | 0.8878 | 0.9178 | 0.945 | 0.105 | 0.9000 | 0.9219 |
| | 2.0 | 0 | 0.920 | 0.010 | 0.9892 | 0.9533 | 0.930 | 0.015 | 0.9841 | 0.9563 |
| | | 1 | 0.990 | 0.030 | 0.9705 | 0.9801 | 1.000 | 0.030 | 0.9708 | 0.9852 |
| | | 3 | 0.990 | 0.065 | 0.9383 | 0.9635 | 0.990 | 0.035 | 0.9658 | 0.9777 |
| | | 5 | 0.975 | 0.110 | 0.8986 | 0.9352 | 0.980 | 0.105 | 0.9032 | 0.9400 |
| | 3.0 | 0 | 0.900 | 0.010 | 0.9890 | 0.9424 | 0.940 | 0.030 | 0.9691 | 0.9543 |
| | | 1 | 0.995 | 0.035 | 0.9660 | 0.9802 | 1.000 | 0.025 | 0.9756 | 0.9876 |
| | | 3 | 1.000 | 0.065 | 0.9389 | 0.9685 | 1.000 | 0.065 | 0.9389 | 0.9685 |
| | | 5 | 1.000 | 0.080 | 0.9259 | 0.9615 | 1.000 | 0.115 | 0.8968 | 0.9456 |
| | 6.0 | 0 | 0.925 | 0.015 | 0.9840 | 0.9536 | 0.930 | 0.010 | 0.9893 | 0.9587 |
| | | 1 | 1.000 | 0.005 | 0.9950 | 0.9975 | 1.000 | 0.015 | 0.9852 | 0.9925 |
| | | 3 | 1.000 | 0.060 | 0.9433 | 0.9708 | 1.000 | 0.090 | 0.9174 | 0.9569 |
| | | 5 | 0.995 | 0.130 | 0.8844 | 0.9364 | 1.000 | 0.110 | 0.9009 | 0.9478 |

Table 15: Detection performance under 10% token-decreasing synonym substitution with OPT-1.3B.

| Setting | | | T200 | | | | T500 | | | |
|---|---|---|---|---|---|---|---|---|---|---|
| Model | $\delta$ | $s_{max}$ | TPR | FPR | Precision | F1 | TPR | FPR | Precision | F1 |
| RS-Watermark | 1.5 | - | 0.940 | 0.925 | 0.5040 | 0.6562 | 0.990 | 0.950 | 0.5103 | 0.6735 |
| | 2 | - | 0.965 | 0.925 | 0.5106 | 0.6678 | 0.970 | 0.940 | 0.5079 | 0.6667 |
| | 3 | - | 0.985 | 0.910 | 0.5198 | 0.6805 | 0.955 | 0.945 | 0.5026 | 0.6586 |
| | 6 | - | 0.980 | 0.915 | 0.5172 | 0.6770 | 0.970 | 0.935 | 0.5092 | 0.6678 |
| Structured-Ours | 1.5 | 0 | 0.750 | 0.005 | 0.9934 | 0.8547 | 0.870 | 0.030 | 0.9647 | 0.8865 |
| | | 1 | 0.900 | 0.020 | 0.9783 | 0.9375 | 0.965 | 0.065 | 0.9369 | 0.9508 |
| | | 3 | 0.905 | 0.055 | 0.9427 | 0.9235 | 0.980 | 0.175 | 0.8485 | 0.9095 |
| | | 5 | 0.910 | 0.100 | 0.9010 | 0.9055 | 0.995 | 0.320 | 0.7567 | 0.8596 |
| | 2.0 | 0 | 0.875 | 0.020 | 0.9777 | 0.9235 | 0.900 | 0.030 | 0.9611 | 0.9326 |
| | | 1 | 0.985 | 0.030 | 0.9704 | 0.9777 | 0.985 | 0.040 | 0.9610 | 0.9728 |
| | | 3 | 0.995 | 0.080 | 0.9256 | 0.9590 | 0.995 | 0.150 | 0.8690 | 0.9277 |
| | | 5 | 0.995 | 0.075 | 0.9299 | 0.9614 | 1.000 | 0.190 | 0.8403 | 0.9132 |
| | 3.0 | 0 | 0.895 | 0.010 | 0.9889 | 0.9396 | 0.935 | 0.020 | 0.9791 | 0.9565 |
| | | 1 | 1.000 | 0.010 | 0.9901 | 0.9950 | 0.995 | 0.075 | 0.9299 | 0.9614 |
| | | 3 | 1.000 | 0.045 | 0.9569 | 0.9780 | 1.000 | 0.160 | 0.8621 | 0.9259 |
| | | 5 | 1.000 | 0.105 | 0.9050 | 0.9501 | 1.000 | 0.280 | 0.7813 | 0.8772 |
| | 6.0 | 0 | 0.905 | 0.020 | 0.9784 | 0.9403 | 0.925 | 0.030 | 0.9686 | 0.9463 |
| | | 1 | 0.990 | 0.025 | 0.9754 | 0.9826 | 0.995 | 0.080 | 0.9256 | 0.9590 |
| | | 3 | 1.000 | 0.070 | 0.9346 | 0.9662 | 1.000 | 0.165 | 0.8584 | 0.9238 |
| | | 5 | 1.000 | 0.125 | 0.8889 | 0.9412 | 1.000 | 0.265 | 0.7905 | 0.8830 |

Table 16: Detection performance under 10% token-deleting synonym substitution using LLaMA-3.2-3B.

| Setting | | | T200 | | | | T500 | | | |
|---|---|---|---|---|---|---|---|---|---|---|
| Model | $\delta$ | $s_{max}$ | TPR | FPR | Precision | F1-score | TPR | FPR | Precision | F1-score |
| RS-Watermark | 1.5 | - | 0.940 | 0.955 | 0.4960 | 0.6493 | 0.930 | 0.930 | 0.5000 | 0.6503 |
| | 2.0 | - | 0.940 | 0.930 | 0.5026 | 0.6551 | 0.925 | 0.950 | 0.4933 | 0.6435 |
| | 3.0 | - | 0.955 | 0.960 | 0.4986 | 0.6552 | 0.945 | 0.925 | 0.5053 | 0.6585 |
| | 6.0 | - | 0.925 | 0.930 | 0.4986 | 0.6479 | 0.945 | 0.930 | 0.5040 | 0.6573 |
| Structured-Ours | 1.5 | 0 | 0.945 | 0.015 | 0.9843 | 0.9642 | 0.830 | 0.025 | 0.9707 | 0.8948 |
| | | 1 | 0.930 | 0.030 | 0.9687 | 0.9489 | 0.900 | 0.065 | 0.9326 | 0.9160 |
| | | 3 | 0.955 | 0.050 | 0.9502 | 0.9526 | 0.950 | 0.060 | 0.9405 | 0.9452 |
| | | 5 | 0.950 | 0.055 | 0.9452 | 0.9476 | 0.905 | 0.075 | 0.9234 | 0.9141 |
| | 2.0 | 0 | 0.900 | 0.015 | 0.9836 | 0.9399 | 0.915 | 0.025 | 0.9734 | 0.9433 |
| | | 1 | 0.965 | 0.050 | 0.9507 | 0.9578 | 0.990 | 0.040 | 0.9611 | 0.9753 |
| | | 3 | 0.970 | 0.090 | 0.9151 | 0.9417 | 0.990 | 0.065 | 0.9383 | 0.9635 |
| | | 5 | 0.995 | 0.120 | 0.8923 | 0.9408 | 0.990 | 0.090 | 0.9166 | 0.9519 |
| | 3.0 | 0 | 0.925 | 0.005 | 0.9946 | 0.9585 | 0.935 | 0.020 | 0.9791 | 0.9565 |
| | | 1 | 0.990 | 0.020 | 0.9801 | 0.9851 | 0.985 | 0.045 | 0.9563 | 0.9704 |
| | | 3 | 0.985 | 0.070 | 0.9336 | 0.9586 | 1.000 | 0.065 | 0.9389 | 0.9685 |
| | | 5 | 1.000 | 0.095 | 0.9132 | 0.9546 | 0.995 | 0.080 | 0.9255 | 0.9590 |
| | 6.0 | 0 | 0.940 | 0.010 | 0.9894 | 0.9641 | 0.915 | 0.010 | 0.9892 | 0.9506 |
| | | 1 | 0.995 | 0.030 | 0.9707 | 0.9827 | 0.995 | 0.030 | 0.9707 | 0.9827 |
| | | 3 | 1.000 | 0.065 | 0.9389 | 0.9685 | 1.000 | 0.045 | 0.9569 | 0.9779 |
| | | 5 | 1.000 | 0.085 | 0.9216 | 0.9592 | 1.000 | 0.115 | 0.8968 | 0.9456 |

Table 17: Detection performance under 5% token-increasing synonym substitution with OPT-1.3B.

| Setting | | | T200 | | | | T500 | | | |
|---|---|---|---|---|---|---|---|---|---|---|
| Model | $\delta$ | $s_{\max}$ | TPR | FPR | Precision | F1 | TPR | FPR | Precision | F1 |
| RS-Watermark | 1.5 | - | 0.945 | 0.925 | 0.5053 | 0.6585 | 0.965 | 0.945 | 0.5052 | 0.6632 |
| | 2 | - | 0.965 | 0.920 | 0.5119 | 0.6669 | 0.950 | 0.960 | 0.4974 | 0.6529 |
| | 3 | - | 0.970 | 0.915 | 0.5146 | 0.6724 | 0.960 | 0.960 | 0.5000 | 0.6575 |
| | 6 | - | 0.960 | 0.920 | 0.5106 | 0.6667 | 0.940 | 0.920 | 0.5034 | 0.6573 |
| Structured-Ours | 1.5 | 0 | 0.245 | 0.010 | 0.9601 | 0.3904 | 0.305 | 0.025 | 0.9242 | 0.4586 |
| | | 1 | 0.380 | 0.025 | 0.9383 | 0.5409 | 0.440 | 0.060 | 0.8800 | 0.5867 |
| | | 3 | 0.525 | 0.050 | 0.9130 | 0.6667 | 0.500 | 0.155 | 0.7634 | 0.6042 |
| | | 5 | 0.635 | 0.075 | 0.8844 | 0.7427 | 0.730 | 0.195 | 0.7892 | 0.7584 |
| | 2.0 | 0 | 0.345 | 0.020 | 0.9452 | 0.5055 | 0.370 | 0.020 | 0.9487 | 0.5324 |
| | | 1 | 0.515 | 0.025 | 0.9537 | 0.6688 | 0.635 | 0.090 | 0.8759 | 0.7362 |
| | | 3 | 0.695 | 0.090 | 0.8854 | 0.7787 | 0.765 | 0.130 | 0.8547 | 0.8074 |
| | | 5 | 0.885 | 0.120 | 0.8806 | 0.8828 | 0.830 | 0.265 | 0.7580 | 0.7924 |
| | 3.0 | 0 | 0.520 | 0.020 | 0.9629 | 0.6753 | 0.475 | 0.020 | 0.9596 | 0.6355 |
| | | 1 | 0.735 | 0.050 | 0.9363 | 0.8235 | 0.695 | 0.115 | 0.8580 | 0.7980 |
| | | 3 | 0.840 | 0.070 | 0.9231 | 0.8796 | 0.920 | 0.185 | 0.8326 | 0.8741 |
| | | 5 | 0.930 | 0.085 | 0.9163 | 0.9231 | 0.945 | 0.250 | 0.7908 | 0.8610 |
| | 6.0 | 0 | 0.600 | 0.020 | 0.9677 | 0.7407 | 0.565 | 0.020 | 0.9658 | 0.7128 |
| | | 1 | 0.755 | 0.020 | 0.9742 | 0.8507 | 0.770 | 0.080 | 0.9059 | 0.8324 |
| | | 3 | 0.950 | 0.080 | 0.9223 | 0.9360 | 0.935 | 0.185 | 0.8348 | 0.8821 |
| | | 5 | 0.935 | 0.100 | 0.9034 | 0.9189 | 0.975 | 0.255 | 0.7927 | 0.8744 |

Table 18: Detection performance under 5% token-inserting synonym substitution using LLaMA-3.2-3B.

| Setting | | | T200 | | | | T500 | | | |
|---|---|---|---|---|---|---|---|---|---|---|
| Model | $\delta$ | $s_{\max}$ | TPR | FPR | Precision | F1-score | TPR | FPR | Precision | F1-score |
| RS-Watermark | 1.5 | - | 0.940 | 0.920 | 0.5053 | 0.6573 | 0.945 | 0.925 | 0.5053 | 0.6585 |
| | 2.0 | - | 0.940 | 0.935 | 0.5013 | 0.6539 | 0.945 | 0.945 | 0.5000 | 0.6539 |
| | 3.0 | - | 0.955 | 0.945 | 0.5026 | 0.6586 | 0.950 | 0.920 | 0.5080 | 0.6620 |
| | 6.0 | - | 0.940 | 0.950 | 0.4973 | 0.6505 | 0.915 | 0.940 | 0.4932 | 0.6409 |
| Structured-Ours | 1.5 | 0 | 0.460 | 0.025 | 0.9484 | 0.6195 | 0.345 | 0.015 | 0.9583 | 0.5073 |
| | | 1 | 0.505 | 0.035 | 0.9351 | 0.6558 | 0.460 | 0.025 | 0.9484 | 0.6195 |
| | | 3 | 0.620 | 0.055 | 0.9185 | 0.7402 | 0.575 | 0.060 | 0.9055 | 0.7033 |
| | | 5 | 0.725 | 0.130 | 0.8479 | 0.7816 | 0.705 | 0.130 | 0.8443 | 0.7683 |
| | 2.0 | 0 | 0.470 | 0.005 | 0.9894 | 0.6372 | 0.445 | 0.040 | 0.9175 | 0.5993 |
| | | 1 | 0.600 | 0.025 | 0.9600 | 0.7384 | 0.565 | 0.010 | 0.9826 | 0.7174 |
| | | 3 | 0.790 | 0.075 | 0.9132 | 0.8471 | 0.715 | 0.100 | 0.8773 | 0.7878 |
| | | 5 | 0.820 | 0.100 | 0.8913 | 0.8541 | 0.810 | 0.110 | 0.8804 | 0.8437 |
| | 3.0 | 0 | 0.515 | 0.015 | 0.9716 | 0.6732 | 0.585 | 0.010 | 0.9832 | 0.7335 |
| | | 1 | 0.675 | 0.025 | 0.9642 | 0.7941 | 0.795 | 0.020 | 0.9754 | 0.8760 |
| | | 3 | 0.865 | 0.060 | 0.9351 | 0.8987 | 0.820 | 0.055 | 0.9371 | 0.8746 |
| | | 5 | 0.935 | 0.075 | 0.9257 | 0.9303 | 0.940 | 0.085 | 0.9171 | 0.9283 |
| | 6.0 | 0 | 0.580 | 0.010 | 0.9831 | 0.7295 | 0.655 | 0.015 | 0.9776 | 0.7844 |
| | | 1 | 0.830 | 0.015 | 0.9822 | 0.8997 | 0.705 | 0.025 | 0.9657 | 0.8150 |
| | | 3 | 0.890 | 0.095 | 0.9035 | 0.8967 | 0.945 | 0.060 | 0.9402 | 0.9426 |
| | | 5 | 0.955 | 0.085 | 0.9182 | 0.9362 | 0.960 | 0.085 | 0.9186 | 0.9388 |

Table 19: Detection performance under 10% token-increasing synonym substitution with OPT-1.3B.

| Setting | | | T200 | | | | T500 | | | |
|---|---|---|---|---|---|---|---|---|---|---|
| Model | $\delta$ | $s_{max}$ | TPR | FPR | Precision | F1 | TPR | FPR | Precision | F1 |
| RS-Watermark | 1.5 | - | 0.970 | 0.975 | 0.4987 | 0.6587 | 0.985 | 0.940 | 0.5117 | 0.6735 |
| | 2 | - | 0.960 | 0.960 | 0.5000 | 0.6575 | 0.980 | 0.945 | 0.5091 | 0.6701 |
| | 3 | - | 0.970 | 0.930 | 0.5105 | 0.6690 | 0.980 | 0.945 | 0.5052 | 0.6632 |
| | 6 | - | 0.985 | 0.935 | 0.5130 | 0.6747 | 0.965 | 0.950 | 0.5039 | 0.6621 |
| Structured-Ours | 1.5 | 0 | 0.150 | 0.025 | 0.8571 | 0.2553 | 0.145 | 0.025 | 0.8529 | 0.2479 |
| | | 1 | 0.165 | 0.040 | 0.8049 | 0.2739 | 0.305 | 0.055 | 0.8472 | 0.4485 |
| | | 3 | 0.305 | 0.050 | 0.8592 | 0.4502 | 0.455 | 0.155 | 0.7459 | 0.5852 |
| | | 5 | 0.360 | 0.085 | 0.8090 | 0.4983 | 0.460 | 0.310 | 0.5974 | 0.5198 |
| | 2.0 | 0 | 0.185 | 0.020 | 0.9024 | 0.3071 | 0.245 | 0.010 | 0.9608 | 0.3904 |
| | | 1 | 0.310 | 0.055 | 0.8493 | 0.4542 | 0.370 | 0.085 | 0.8132 | 0.5086 |
| | | 3 | 0.380 | 0.080 | 0.8261 | 0.5205 | 0.580 | 0.160 | 0.7838 | 0.6667 |
| | | 5 | 0.480 | 0.115 | 0.8067 | 0.6019 | 0.645 | 0.305 | 0.6789 | 0.6615 |
| | 3.0 | 0 | 0.325 | 0.010 | 0.9701 | 0.4869 | 0.355 | 0.015 | 0.9595 | 0.5182 |
| | | 1 | 0.505 | 0.020 | 0.9619 | 0.6623 | 0.530 | 0.075 | 0.8760 | 0.6604 |
| | | 3 | 0.645 | 0.070 | 0.9021 | 0.7522 | 0.755 | 0.135 | 0.8483 | 0.7989 |
| | | 5 | 0.710 | 0.110 | 0.8659 | 0.7802 | 0.865 | 0.260 | 0.7689 | 0.8141 |
| | 6.0 | 0 | 0.355 | 0.005 | 0.9861 | 0.5221 | 0.430 | 0.015 | 0.9663 | 0.5952 |
| | | 1 | 0.500 | 0.035 | 0.9346 | 0.6515 | 0.630 | 0.075 | 0.8936 | 0.7390 |
| | | 3 | 0.730 | 0.050 | 0.9359 | 0.8202 | 0.815 | 0.105 | 0.8859 | 0.8490 |
| | | 5 | 0.815 | 0.110 | 0.8810 | 0.8468 | 0.960 | 0.265 | 0.7837 | 0.8629 |

Table 20: Detection performance under 10% token-inserting synonym substitution using LLaMA-3.2-3B.

| Setting | | | T200 | | | | T500 | | | |
|---|---|---|---|---|---|---|---|---|---|---|
| Model | $\delta$ | $s_{max}$ | TPR | FPR | Precision | F1-score | TPR | FPR | Precision | F1-score |
| RS-Watermark | 1.5 | - | 0.920 | 0.940 | 0.4946 | 0.6433 | 0.920 | 0.940 | 0.4946 | 0.6433 |
| | 2.0 | - | 0.920 | 0.935 | 0.4960 | 0.6444 | 0.945 | 0.955 | 0.4973 | 0.6517 |
| | 3.0 | - | 0.915 | 0.975 | 0.4841 | 0.6332 | 0.930 | 0.905 | 0.5068 | 0.6561 |
| | 6.0 | - | 0.915 | 0.930 | 0.4959 | 0.6432 | 0.935 | 0.945 | 0.4973 | 0.6493 |
| Structured-Ours | 1.5 | 0 | 0.215 | 0.020 | 0.9148 | 0.3481 | 0.210 | 0.000 | 1.0000 | 0.3471 |
| | | 1 | 0.195 | 0.020 | 0.9069 | 0.3209 | 0.265 | 0.040 | 0.8688 | 0.4061 |
| | | 3 | 0.370 | 0.070 | 0.8409 | 0.5138 | 0.350 | 0.065 | 0.8433 | 0.4946 |
| | | 5 | 0.425 | 0.060 | 0.8762 | 0.5723 | 0.430 | 0.105 | 0.8037 | 0.5602 |
| | 2.0 | 0 | 0.315 | 0.020 | 0.9402 | 0.4719 | 0.245 | 0.040 | 0.8596 | 0.3813 |
| | | 1 | 0.335 | 0.030 | 0.9294 | 0.5543 | 0.340 | 0.035 | 0.9066 | 0.4945 |
| | | 3 | 0.480 | 0.050 | 0.9056 | 0.6274 | 0.495 | 0.105 | 0.8250 | 0.6187 |
| | | 5 | 0.565 | 0.100 | 0.8496 | 0.6786 | 0.520 | 0.065 | 0.8888 | 0.6561 |
| | 3.0 | 0 | 0.370 | 0.010 | 0.9736 | 0.5362 | 0.430 | 0.005 | 0.9885 | 0.5993 |
| | | 1 | 0.420 | 0.035 | 0.9231 | 0.5773 | 0.420 | 0.025 | 0.9438 | 0.5813 |
| | | 3 | 0.565 | 0.070 | 0.8897 | 0.6911 | 0.575 | 0.060 | 0.9055 | 0.7033 |
| | | 5 | 0.690 | 0.130 | 0.8414 | 0.7582 | 0.695 | 0.100 | 0.8742 | 0.7743 |
| | 6.0 | 0 | 0.390 | 0.000 | 1.0000 | 0.5611 | 0.370 | 0.015 | 0.9610 | 0.5443 |
| | | 1 | 0.515 | 0.040 | 0.9279 | 0.6623 | 0.485 | 0.045 | 0.9151 | 0.6339 |
| | | 3 | 0.720 | 0.075 | 0.9056 | 0.8022 | 0.660 | 0.050 | 0.9295 | 0.7719 |
| | | 5 | 0.765 | 0.095 | 0.8895 | 0.8225 | 0.800 | 0.120 | 0.8695 | 0.8333 |

Table 21: Detection performance of Structured-Ours and RS-Watermark under a 20% mixed synonym substitution attack (combining insertion-like, deletion-like, and replacement-like effects) on the C4 dataset using OPT-1.3B.Structured-Ours maintains consistently low FPR, whereas RS-Watermark exhibits extremely high FPR despite high TPR.

| Setting | | | T200 | | | | T500 | | | |
| --- | --- | --- | --- | --- | --- | --- | --- | --- | --- | --- |
| Model | $\delta$ | $s_{max}$ | TPR | FPR | Precision | F1 score | TPR | FPR | Precision | F1 score |
| RS-Watermark | 1.5 | - | 0.9650 | 0.9500 | 0.5039 | 0.6621 | 0.9400 | 0.9700 | 0.4921 | 0.6460 |
| | 2 | - | 0.9200 | 0.9600 | 0.4894 | 0.6389 | 0.9400 | 0.9500 | 0.4974 | 0.6505 |
| | 3 | - | 0.9300 | 0.9600 | 0.4921 | 0.6436 | 0.9850 | 0.9100 | 0.5198 | 0.6805 |
| | 6 | - | 0.9600 | 0.9050 | 0.5147 | 0.6702 | 0.9850 | 0.9350 | 0.5131 | 0.6747 |
| Structured-Our | 1.5 | 0 | 0.0150 | 0.0000 | 1.0000 | 0.0296 | 0.0700 | 0.0300 | 0.7000 | 0.1273 |
| | | 1 | 0.0700 | 0.0200 | 0.7778 | 0.1284 | 0.1250 | 0.1000 | 0.5556 | 0.2041 |
| | | 3 | 0.1100 | 0.0550 | 0.6667 | 0.1888 | 0.2050 | 0.1550 | 0.5694 | 0.3015 |
| | | 5 | 0.1150 | 0.0500 | 0.6970 | 0.1974 | 0.3100 | 0.2150 | 0.5905 | 0.4066 |
| | 2 | 0 | 0.0150 | 0.0100 | 0.6000 | 0.0293 | 0.0300 | 0.0150 | 0.6667 | 0.0574 |
| | | 1 | 0.0600 | 0.0300 | 0.6667 | 0.1101 | 0.1400 | 0.0500 | 0.7368 | 0.2353 |
| | | 3 | 0.1400 | 0.0550 | 0.7179 | 0.2343 | 0.2100 | 0.1800 | 0.5385 | 0.3022 |
| | | 5 | 0.1700 | 0.0900 | 0.6538 | 0.2698 | 0.3400 | 0.1750 | 0.6602 | 0.4488 |
| | 3 | 0 | 0.0700 | 0.0150 | 0.8235 | 0.1290 | 0.0800 | 0.0250 | 0.7619 | 0.1448 |
| | | 1 | 0.1000 | 0.0200 | 0.8333 | 0.1786 | 0.1250 | 0.0850 | 0.5952 | 0.2066 |
| | | 3 | 0.1800 | 0.0700 | 0.7200 | 0.2880 | 0.3400 | 0.1750 | 0.6602 | 0.4498 |
| | | 5 | 0.2100 | 0.1000 | 0.6774 | 0.3206 | 0.3900 | 0.2800 | 0.5821 | 0.4671 |
| | 6 | 0 | 0.1150 | 0.0150 | 0.8846 | 0.2035 | 0.0750 | 0.0300 | 0.7143 | 0.1357 |
| | | 1 | 0.1450 | 0.0350 | 0.8056 | 0.2458 | 0.2700 | 0.0450 | 0.8571 | 0.4106 |
| | | 3 | 0.3300 | 0.0800 | 0.8049 | 0.4681 | 0.4350 | 0.1700 | 0.7190 | 0.5421 |
| | | 5 | 0.3650 | 0.1500 | 0.7087 | 0.4818 | 0.5600 | 0.2000 | 0.7368 | 0.6364 |

Table 22: Detection performance under the same 20% mixed synonym substitution attack on the OpenGen dataset. Structured-Ours again maintains low FPR across settings, while RS-Watermark shows near-random FPR across $\delta$ values.

| Setting | | | T200 | | | | T500 | | | |
| --- | --- | --- | --- | --- | --- | --- | --- | --- | --- | --- |
| Model | $\delta$ | $s_{max}$ | TPR | FPR | Precision | F1 score | TPR | FPR | Precision | F1 score |
| RS-Watermark | 1.5 | - | 0.9150 | 0.9600 | 0.4880 | 0.6365 | 0.9600 | 0.9750 | 0.4961 | 0.6542 |
| | 2 | - | 0.9400 | 0.9300 | 0.5027 | 0.6551 | 0.9350 | 0.9300 | 0.5013 | 0.6527 |
| | 3 | - | 0.9300 | 0.9200 | 0.5027 | 0.6526 | 0.9700 | 0.9200 | 0.5132 | 0.6713 |
| | 6 | - | 0.9300 | 0.9350 | 0.4987 | 0.6492 | 0.9900 | 0.9450 | 0.5116 | 0.6746 |
| Structured-Our | 1.5 | 0 | 0.0350 | 0.0050 | 0.8750 | 0.0673 | 0.0300 | 0.0150 | 0.6667 | 0.0574 |
| | | 1 | 0.0700 | 0.0350 | 0.6667 | 0.1267 | 0.1150 | 0.0900 | 0.5610 | 0.1909 |
| | | 3 | 0.0650 | 0.0600 | 0.5200 | 0.1156 | 0.2000 | 0.1650 | 0.5479 | 0.2930 |
| | | 5 | 0.1300 | 0.1150 | 0.5306 | 0.2088 | 0.3100 | 0.1950 | 0.6139 | 0.4120 |
| | 2 | 0 | 0.0300 | 0.0100 | 0.7500 | 0.0577 | 0.0350 | 0.0350 | 0.5000 | 0.0654 |
| | | 1 | 0.0450 | 0.0150 | 0.7500 | 0.0849 | 0.1750 | 0.0750 | 0.7000 | 0.2800 |
| | | 3 | 0.1300 | 0.0700 | 0.6500 | 0.2167 | 0.2600 | 0.1700 | 0.6047 | 0.3636 |
| | | 5 | 0.2100 | 0.1150 | 0.6462 | 0.3170 | 0.3350 | 0.2250 | 0.5982 | 0.4295 |
| | 3 | 0 | 0.0650 | 0.0050 | 0.9286 | 0.1215 | 0.0750 | 0.0250 | 0.7500 | 0.1364 |
| | | 1 | 0.1150 | 0.0150 | 0.8846 | 0.2035 | 0.1600 | 0.0850 | 0.6531 | 0.2570 |
| | | 3 | 0.1600 | 0.0800 | 0.6667 | 0.2581 | 0.3000 | 0.1300 | 0.6977 | 0.4196 |
| | | 5 | 0.2200 | 0.1200 | 0.6471 | 0.3284 | 0.4700 | 0.2650 | 0.6395 | 0.5418 |
| | 6 | 0 | 0.0850 | 0.0050 | 0.9444 | 0.1560 | 0.1300 | 0.0400 | 0.7647 | 0.2222 |
| | | 1 | 0.1400 | 0.0450 | 0.7568 | 0.2363 | 0.3050 | 0.0650 | 0.8243 | 0.4453 |
| | | 3 | 0.3300 | 0.0850 | 0.7952 | 0.4664 | 0.4100 | 0.1600 | 0.7193 | 0.5223 |
| | | 5 | 0.3500 | 0.1150 | 0.7527 | 0.4778 | 0.6050 | 0.2600 | 0.6994 | 0.6488 |

