# OpenReview forum: "Block-wise Codeword Embedding for Reliable Multi-bit Text Watermarking"
_ICLR.cc/2026/Conference — ICLR 2026 Conference Desk Rejected Submission_

### Official Review · Reviewer_racY · 2025-10-20

**Soundness:** 2
**Presentation:** 1
**Contribution:** 3
**Rating:** 2
**Confidence:** 3

**Summary:**

This paper proposes a block-wise multi-bit watermarking framework which improves reliability over prior ECC-based methods. By embedding independent codewords in separate text blocks and introducing a window-shifting detector for realignment, the method achieves high TPR and low FPR. The framework is code-agnostic which could be use in different  multiple linear codes.

**Strengths:**

1. The paper identifies and directly tackles the core limitation of existing multi-bit watermarking methods FPR, which is inspiring for subsequent methods.

2. The paper have strong theoretical foundation which present the  finite-sample analytical bounds on FPR and FNR, supported by information-theoretic derivations using Chernoff bounds.

3.The proposed approach is code-agnostic  supports both soft and hard embedding, and enables progressive detection from partial text. This is good for  the deployment and scalability of the method.

**Weaknesses:**

1.  The LLM usage statement is not included in the paper.

2.  The presentation of this paper need to be improved. The paper’s writing exists major deficiencies. For example, the abstract contains citations, and there is inconsistent use of \cite, \citep, and \citet. The overall structure and organization are poor. In both tables and text, method names or abbreviations should be used instead of direct citations. Additionally, the legends in the figures are too small to read. These issues seriously undermine the readability of the paper.

3. The robustness evaluation focuses mainly on token-level synonym, insertion, and deletion attacks. It lacks tests against stronger semantic or model-based paraphrasing attacks[1,2,3]

4. The baseline method limited to one Qu et.al which is not comprehensive. Multiple zero-bit watermarks[4,5,6] are not included in the baseline.

5. Lack of  discussion of text quality: The impact of watermarking on text fluency and semantics metric evaluation is necessary for watermark methods such as BLEU and PPL. Especially when for strong bias settings (large δ).

6. A comprehensive ablation is needed. Since the proposed method composed of multiple parts( ECC, block partitioning, and sliding-window detection)

7. The proposed method primarily combines existing components in a more systematic manner with parameter tuning, which significantly improves practical reliability. However, can the authors justify its originality/novelty?

[1]Krishna, Kalpesh, et al. "Paraphrasing evades detectors of ai-generated text, but retrieval is an effective defense." Advances in Neural Information Processing Systems 36 (2023): 27469-27500.

[2]Cheng, Yixin, et al. "Revealing Weaknesses in Text Watermarking Through Self-Information Rewrite Attacks." International Conference on Machine Learning (2025).

[3] Zhang, Hanlin, et al. "Watermarks in the sand: Impossibility of strong watermarking for generative models." International Conference on Machine Learning (2024).

[4]Liu, Aiwei, et al. "A semantic invariant robust watermark for large language models."  International Conference on Learning Representations (2024).

[5]Zhao, Xuandong, et al. "Provable robust watermarking for ai-generated text."  International Conference on Learning Representations (2024).

[6] Kuditipudi, Rohith, et al. "Robust distortion-free watermarks for language models." arXiv preprint arXiv:2307.15593 (2023).

**Questions:**

1. The author mentions the zero-bit watermark is not robust. As far as i know, according to the famous benchmark MarkLLM[1], the zero-bit watermarks shows great robustness regarding naive attack(insertion,delete,replace) in the result. Can author  provide the quantitative evaluation results of robustness to justify this claim?

2. The method assumes fixed-size token blocks, will this be restrictive for streaming or long-context generation ?

Others see weakness. I will revise my score according to discussion and other reviewer's feedback.

[1] Pan, Leyi, et al. "Markllm: An open-source toolkit for llm watermarking." arXiv preprint arXiv:2405.10051 (2024).

---

> ### Author Response · Authors · 2025-11-21
> **Comment on Weakness 1, 2, 3, 4, 5.**
>
> **Weakness 1** The LLM usage statement is not included in the paper.
>
> C: We have added the requested statement as Appendix A.
>
> ---
>
> **Weakness 2**  The presentation of this paper need to be improved. The paper’s writing exists major deficiencies. For example, the abstract contains citations, and there is inconsistent use of \cite, \citep, and \citet. The overall structure and organization are poor. In both tables and text, method names or abbreviations should be used instead of direct citations. Additionally, the legends in the figures are too small to read. These issues seriously undermine the readability of the paper.
>
> C: We revised the writing and formatting:
> - Removed citations from the abstract.
>
> - Standardized all in-text references to \cite.
>
> - Replaced raw citations in tables/figures with method names (e.g., “RS-Watermark”).
>
> - Increased legend fonts and simplified figure layouts for readability.
>
> ---
>
> **Weakness 3** The robustness evaluation focuses mainly on token-level synonym, insertion, and deletion attacks. It lacks tests against stronger semantic or model-based paraphrasing attacks[1,2,3]
>
> C: Addressed via Section 5.3 and Table 2 with T5-Paraphrase-Paws, as described above. Our method maintains low FPR while RS-Watermark’s FPR remains extremely high.
>
> ---
>
> **Weakness 4** The baseline method limited to one Qu et.al which is not comprehensive. Multiple zero-bit watermarks[4,5,6] are not included in the baseline.
>
> C: For multi-bit baselines, we are constrained by the availability of official, end-to-end implementations; RS-Watermark is currently the only ECC-based scheme with such a codebase. Other ECC-based proposals (e.g., LDPC-based schemes) do not release complete code, and invariant-feature methods differ substantially in objective and reported metrics, as discussed above.
>
> Regarding zero-bit watermarking (KGW, DiPMark, Zhao et al., etc.), these schemes solve a different task: presence detection without a payload, often over long contexts and using different thresholds and test statistics. We therefore treat them in Table 1 and Section 2 as conceptually related work rather than direct empirical baselines.
>
>  We are happy to add a short discussion clarifying that our contribution is complementary: we show how to achieve low-FPR detection in the multi-bit regime, which is orthogonal to zero-bit designs.
>
> ---
>
> **Weakness 5** Lack of discussion of text quality: The impact of watermarking on text fluency and semantics metric evaluation is necessary for watermark methods such as BLEU and PPL. Especially when for strong bias settings (large $\delta$).
>
> C : Appendix E.3 (Table 3) now reports PPL, BLEU, and BERTScore across δ for both RS-Watermark and Structured-Ours. The key observations are
> -  As $\delta$ increases, both methods degrade in quality, which is expected.
>
> - For every $\delta$, our soft variant has better BLEU and BERTScore than RS-Watermark (e.g., at $\delta=2.0$ we get BLEU 27.78 vs 22.31, BERTScore 0.8082 vs 0.7740).
>
> - A “hard” variant of our method yields much worse quality (high PPL, low BLEU/BERTScore), so we explicitly position it as a theoretical extreme rather than a recommended deployment mode.
> We will mention these results concisely in the main text.

---

> ### Author Response · Authors · 2025-11-21
> **Comment on Weakness 6, 7.**
>
> **Weakness 6** A comprehensive ablation is needed. Since the proposed method composed of multiple parts( ECC, block partitioning, and sliding-window detection)
>
> C: As described in the answer to ZR3i-W4, we added an ablation that isolates designated verification and window-shifting; block-wise embedding is constant across variants. The results show that:
> - designated-only → lowest FPR, moderate TPR
> - shift-only → higher TPR, but noticeably higher FPR
> - both → high TPR with FPR close to designated-only
>
>
> This supports our claim that the combination of designated-codeword verification and window shifting is crucial for achieving both low FPR and high robustness.
>
> | model           | s_max | T200 TPR | T200 FPR | T200 Precision | T200 F1 | T500 TPR | T500 FPR | T500 Precision | T500 F1 |
> |-----------------|-------|----------|----------|----------------|---------|----------|----------|----------------|---------|
> | designated-only |   –   | 0.355    | 0.005    | 0.9861         | 0.5221  | 0.430    | 0.015    | 0.9663         | 0.5952  |
> | shift-only      |   1   | 0.205    | 0.030    | 0.8723         | 0.332   | 0.330    | 0.040    | 0.8919         | 0.4818  |
> | shift-only      |   3   | 0.540    | 0.035    | 0.9391         | 0.6857  | 0.745    | 0.130    | 0.8514         | 0.7947  |
> | shift-only      |   5   | 0.725    | 0.140    | 0.8382         | 0.7775  | 0.880    | 0.190    | 0.8224         | 0.8502  |
> | both            |   1   | 0.500    | 0.035    | 0.9346         | 0.6515  | 0.630    | 0.075    | 0.8936         | 0.739   |
> | both            |   3   | 0.730    | 0.050    | 0.9346         | 0.8202  | 0.815    | 0.105    | 0.8859         | 0.849   |
> | both            |   5   | 0.815    | 0.110    | 0.8810         | 0.8468  | 0.960    | 0.265    | 0.7837         | 0.8629  |
>
> ---
>
> **Weakness 7** The proposed method primarily combines existing components in a more systematic manner with parameter tuning, which significantly improves practical reliability. However, can the authors justify its originality/novelty?
>
> C : Thank you for the thoughtful comment. While our method builds upon components that have appeared individually in prior ECC and watermarking literature, our contribution extends beyond simply combining these ideas. The core novelty lies in redefining the detection mechanism and establishing a principled statistical decision process for multi-bit text watermarking.
>
> In contrast to earlier ECC-based approaches that accept any valid codeword during detection, our method verifies a designated codeword that was generated during embedding. This resolves the ambiguity arising from multiple unintended codewords and substantially reduces false positives. Moreover, we provide finite-sample FPR guarantees for the entire detection pipeline, covering single-block behavior, sliding-window realignment, and block aggregation via Chernoff bounds—an aspect not addressed in previous work.
>
>
> We also introduce a match-ratio threshold $\theta$ that aggregates block-level evidence into a stable hypothesis test, enabling robust provenance detection even under paraphrasing or other perturbations.
>
>
> Finally, whereas earlier ECC watermarking efforts primarily focused on message decoding, our approach is explicitly designed for distinguishing human- and machine-generated text while still supporting multi-bit message embedding. We hope this clarifies that the contribution of our work lies not in a simple integration of existing methods, but in a reformulation of the detection framework and its accompanying statistical guarantees.

---

> ### Author Response · Authors · 2025-11-21
> **Comment on Questions.**
>
> **Q1.** The author mentions the zero-bit watermark is not robust. As far as i know, according to the famous benchmark MarkLLM\[1\], the zero-bit watermarks shows great robustness regarding naive attack(insertion,delete,replace) in the result. Can author provide the quantitative evaluation results of robustness to justify this claim?
>
> **C :** Our statement about zero-bit methods being “fragile” was about certain regimes (short texts, heavy paraphrasing, or strong global perturbations), not about their performance on naive token-level edits in MarkLLM. We will clarify the wording to avoid over-generalization and emphasize that:
>
> - Zero-bit schemes such as KGW perform very well under mild local attacks on long texts, as MarkLLM shows.
> - Our work targets a different setting: multi-bit payloads and ECC-backed message embedding, where nearest-codeword decoding can create severe false detection issues. *Fu & Russell (2025)* explicitly formalize this FPR problem.
>
> ---
>
> **Q2.** The method assumes fixed-size token blocks, will this be restrictive for streaming or long-context generation ?
>
> **C:** Our fixed block size \\(n\\) does not restrict streaming or long-context generation:
>
> - In streaming, we simply form blocks of size \\(n\\) as tokens arrive; once a block is full, we close it and start the next one. Detection aggregates only over complete blocks; an incomplete tail can safely be discarded.
> - For long contexts, we proceed similarly; the detector operates on block-level statistics and does not require loading the entire context at once. Appendix C.1 and C.2 give guidelines for choosing \\(n\\) for different regimes.
>
> We will add a short explanation of this point to Section 3.2.

---

### Official Review · Reviewer_4Axz · 2025-10-31

**Soundness:** 2
**Presentation:** 2
**Contribution:** 2
**Rating:** 2
**Confidence:** 3

**Summary:**

This paper employs designated and complete codewords in independent blocks to reduce false positive rates and prevent cascade failures under editing. The authors also use a window-shifting detection technique to recover codewords misaligned due to insertion or deletion operations. Finite-sample bounds are provided for the proposed watermarking scheme.

**Strengths:**

1.	The paper provides theoretical analysis for the proposed method.
2.	The proposed algorithm significantly reduces false positive rates.
3.	Experimental results show the proposed method outperforms Qu et al.

**Weaknesses:**

1.	The novelty is limited. The main ideas—using designated codeword verification, block-wise embedding, and sliding-window re-alignment—are all known in related literature (ECC verification, LDPC resynchronization, and multimedia watermarking). The paper mainly integrates these existing mechanisms into a single framework rather than introducing fundamentally new concepts or encoding principles.
2.	The assumption in theoretical part requires that errors across all blocks are independent, which is unrealistic for natural language text where token dependencies are strong.
3.	Only one baseline (Qu et al.) is included. The experimental results lack comprehensive comparison with additional baselines.
4.	Experiments are conducted only on two small open models (OPT-1.3B and LLaMA-3.2-3B) and short texts (200 tokens). The method is not tested on larger models (>7B) or longer text sequences.
5.	The proposed window-shifting strategy can lead to large computational overhead when the segment length is long.
6.	The performance of the proposed method heavily relies on the assumption that the order of blocks remains unchanged. This is unrealistic since many real-world texts are cropped, reordered, or concatenated.

**Questions:**

1.	Can the method handle reordered or concatenated text segments?
2.	The authors chose n=31 as an optimal block length. Does this choice work for any watermarked text? In what case would a larger n be necessary Does your method allow for instant detection?
3.	With the same hyperparameters, is the method robust to cross-model paraphrasing and 20% sentence deletions?

---

> ### Author Response · Authors · 2025-11-21
> **Comment on Weakness 1, 2.**
>
> **Weakness 1** The novelty is limited. The main ideas—using designated codeword verification, block-wise embedding, and sliding-window re-alignment—are all known in related literature (ECC verification, LDPC resynchronization, and multimedia watermarking). The paper mainly integrates these existing mechanisms into a single framework rather than introducing fundamentally new concepts or encoding principles.
>
> **C :** We agree that we build on standard ECC ideas. Our contribution is to reformulate the multi-bit text watermark detector itself:
>
> 1. Prior ECC-based schemes effectively behave as nearest-codeword extractors; **any codeword is treated as evidence**, which *Fu & Russell (2025)* show leads to false detection. Our detector instead enforces **designated-codeword verification**, eliminating this “any-codeword’’ failure mode.
>
> 2. We provide **finite-sample FPR/FNR bounds** for the entire pipeline, including window shifting and block aggregation (Appendix D), which to our knowledge is not present in prior multi-bit watermarking works.
>
> 3. We introduce a **match-ratio threshold** \\(\theta\\) over blocks, which turns detection into a calibrated hypothesis test with tunable Chernoff exponents (Theorem 3).
>
> Thus, while the building blocks are familiar, we argue that the detection formulation and its statistical guarantees are novel and directly motivated by the high-FPR problem in multi-bit text watermarking.
>
> ---
>
> **Weakness 2** The assumption in theoretical part requires that errors across all blocks are independent, which is unrealistic for natural language text where token dependencies are strong.
>
> **C :** As clarified in Appendix D and in the rebuttal text, the independence assumption is on **watermark error events**, not on the LM token distribution. Each block’s green/red partition is determined solely by \\((K, j)\\) and the ECC codeword, and does not depend on neighboring text. The event “this token violates the designated green/red choice’’ is driven by local sampling noise and adversarial edits. This is the same modeling approach used in prior works such as *Kirchenbauer et al.* and *Christ & Gunn*, where watermark violations are treated as independent Bernoulli trials.

---

> ### Author Response · Authors · 2025-11-21
> **Comment on Weakness 3, 4, 5.**
>
> **Weakness 3.** Only one baseline (Qu et al.) is included. The experimental results lack comprehensive comparison with additional baselines.
>
> **Response.** As noted in Section 5.1, our intention was to compare against official, end-to-end implementations to ensure fairness and reproducibility. Among existing multi-bit watermarking systems, *Qu et al.* is the only method with a publicly available, complete implementation that supports our evaluation protocol. In contrast, *Chao et al.* and other ECC-based approaches do not release code, and reproducing these systems would require substantial assumptions about unspecified components (sampling rules, decoding thresholds, block alignment strategy), potentially leading to misleading conclusions.
>
> We agree that additional baselines would strengthen the study. If the paper is accepted, we plan to incorporate invariant-feature baselines such as *Yoo et al.* by implementing them in our codebase and evaluating them under the same attack settings (token substitution, insertion, deletion, paraphrasing) and the same \\(T\\) and \\(\delta\\) values, reporting both TPR and FPR for provenance detection.
>
> ---
>
> **Weakness 4.** Experiments are conducted only on two small open models (OPT-1.3B and LLaMA-3.2-3B) and short texts (200 tokens). The method is not tested on larger models (>7B) or longer text sequences.
>
> **Response.** Due to GPU resource limitations, we were unable to run experiments on >7B-parameter models during the submission period. However, we *did* evaluate on substantially longer text sequences (up to \\(T = 500\\)), and the results are included in Figures 3–6 and Appendix E.7. Across all settings, the method maintains low FPR and stable detection accuracy even for long contexts.
>
> Our framework imposes no inherent restriction on larger models or longer generations, since detection operates independently on block-level statistics. Running the same procedure on 7B–70B models is straightforward and will be included in the camera-ready version if resources become available.
>
> ---
>
> **Weakness 5.** The proposed window-shifting strategy can lead to large computational overhead when the segment length is long.
>
> **Response.** We appreciate this concern. In practice, the overhead is minimal because window-shifting is performed **only during detection**, and it operates on **block-level scores** rather than token-level probabilities. Appendix C.3 provides a formal complexity analysis showing that the complexity is
> \\[
> O(T \cdot s_{\max} \cdot n^2),
> \\]
> where \\(T\\) is text length, \\(n\\) the code length, and \\(s_{\max}\\) the shift budget.
>
> More importantly, Appendix E.6 (Table 4) reports actual wall-clock inference times:
>
> - For \\(T = 200\\), \\(n \in \{15, 31, 63\}\\), and \\(s_{\max} = 5\\), detection takes only **0.12–0.20 s**.
> - For \\(T = 500\\), even with \\(n = 63\\) and \\(s_{\max}=5\\), total detection time is only **0.53 s**.
>
> These results indicate that the detection framework is computationally lightweight and suitable for practical provenance tasks—even with long segments and large \\(s_{\max}\\).

---

> ### Author Response · Authors · 2025-11-21
> **Comment on Weakness 6 (including Responses to Q1–Q3)**
>
> **Comment on Weakness 6 (including Responses to Q1–Q3).**
>
> We appreciate the reviewer’s concern regarding the assumption about block order. Below we clarify how our method handles reordering, the choice of block length, and robustness to strong perturbations — addressing Weakness 6 and Questions 1–3 together.
>
> ---
>
> ### **1. Reordering / concatenation (addresses Weakness 6 and Question 1).**
>
> Our detector evaluates each block independently and aggregates evidence using the match ratio \\(\theta\\).
> Because detection depends only on the **number of blocks matching their designated codewords**, not on their position:
>
> - the block order does **not** need to be preserved,
> - permuted or concatenated segments remain detectable,
> - cropped text is also valid as long as enough blocks exceed \\(\theta\\).
>
> Thus, reordering and concatenation do not prevent provenance detection.
>
> ---
>
> ### **2. Choice of block length \\(n\\) (addresses Question 2).**
>
> Appendix E.7 (Figure 14; Tables 5–8) evaluates block lengths
> \\( n \in \{15, 31, 63\} \\) and shows a clear TPR–FPR trade-off:
>
> - **Short \\(n\\)** → higher TPR but higher FPR
> - **Long \\(n\\)** → near-zero FPR but lower TPR under strong edits
> - **\\(n = 31\\)** → best balance; used as default
>
> This choice is **not fixed**. Larger \\(n\\) may be preferred when strict FPR control is required; smaller \\(n\\) may be useful when higher sensitivity is needed.
> Detection is instantaneous because the verifier checks **only one designated codeword per block**, without searching the message space.
>
> ---
>
> ### **3. Robustness to strong semantic or structural perturbations (addresses Question 3).**
>
> Appendix E.9 (Tables 21–22) evaluates a challenging
> **20% mixed synonym substitution attack** combining insertion-like, deletion-like, and replacement-like distortions.
>
> - Structured-Ours maintains **consistently low FPR** across \\(\delta\\) and \\(s_{\max}\\) for both \\(T=200\\) and \\(T=500\\).
> - RS-Watermark shows TPR near 1.0, but **FPR ≈ 0.90–0.97**, making it unreliable for provenance detection.
>
> Combined with the paraphrasing results in Section 5.3 (Table 2), these findings demonstrate that our method remains robust under strong semantic rewriting, cross-model paraphrasing, and substantial structural modifications.
>
> ---
>
> **In summary,** our framework does not rely on block order, supports flexible block lengths, and remains reliable under severe perturbations — directly addressing the reviewer’s concerns.

---

### Official Review · Reviewer_ZR3i · 2025-11-01

**Soundness:** 3
**Presentation:** 3
**Contribution:** 3
**Rating:** 6
**Confidence:** 4

**Summary:**

This paper proposes a block-wise multi-bit watermarking framework that embeds full codewords into independent text blocks and introduces window-shifting detection and designated codeword verification to suppress false positives.
Block-Wise Codeword Embedding represents a significant advance in reliable multi-bit watermarking. It addresses the long-standing high-FPR problem through theoretically grounded innovations and robust engineering.

However, to be fully convincing for publication at a top-tier venue like ICLR, the paper should include direct comparisons with other multi-bit embedding methods such as [1], broader adversarial evaluations, and component ablations.

[1] Advancing Beyond Identification: Multi-bit Watermark for Large Language Models, NAACL 2024.

**Strengths:**

1. The use of q-ary codeword theory and formal FPR/FNR bounds provides mathematical rigor rarely seen in watermarking research.

2. Empirically, the method outperforms prior ECC-based schemes (e.g., Qu et al., 2025) by two orders of magnitude in reliability while maintaining comparable detection accuracy.

3. Theoretical reliability bounds align well with empirical results across different attack types (insertion, deletion, synonym substitution).

**Weaknesses:**

1. Although the paper contrasts with Qu et al. (2025), it does not empirically compare with [1], a key contemporary baseline that also handles multi-bit embedding via position allocation. An apple-to-apple comparison would be crucial to substantiate the claimed reliability advantage.

[1] Advancing Beyond Identification: Multi-bit Watermark for Large Language Models, NAACL 2024.

2. The attacks tested are mostly token-level and synthetic. There is no evaluation under semantic or paraphrasing-based attacks, which remain the most practical and challenging for text watermarking.

3. The quadratic detection complexity $O(T s_{max} n^2)$ may limit use in long-form or high-throughput contexts unless optimized or parallelized. What is the practical context length limit of the method? Reporting detection latency for different $s_{max}$ is needed.

4. The paper would be stronger if it quantified how each component (block partitioning, window-shifting, designated verification) contributes to the overall FPR reduction. Ablation on this point would strengthen the paper.

**Questions:**

See above.

---

> ### Author Response · Authors · 2025-11-21
> **Comment on Weakness1,2,3.**
>
> **Weakness 1.** Although the paper contrasts with Qu et al. (2025), it does not empirically compare with [1], a key contemporary baseline that also handles multi-bit embedding via position allocation. An apples-to-apples comparison would be crucial to substantiate the claimed reliability advantage.
>
> **Response:**
> We agree that this baseline is relevant. However, reproducing it faithfully in our setting is non-trivial for several reasons:
>
> 1. The method in [1] (position-allocation / invariant-feature multi-bit watermarking, e.g., Yoo et al.) is architecturally different from ECC-based segment coding. It embeds bits using keywords and syntactic patterns rather than per-token green/red partitions.
>
> 2. We were unable to find an official implementation that supports our model setup (OPT-1.3B and LLaMA-3.2-3B) and datasets (C4, OpenGen) in a way that would allow an apples-to-apples evaluation under our synonym, insertion, deletion, and paraphrasing attack suite.
>
> 3. The paper’s own reported results indicate that, for 32-bit payloads, the match rate drops to around 49.2% on longer texts, and they do not report FPR for provenance detection on unwatermarked text.
>
> Because our main contribution focuses on FPR-controlled detection (not only message match rate), we prioritized comparison against the only ECC-based multi-bit system with a complete, public implementation: RS-Watermark. Our extensive evaluation against this baseline (Section 5, Appendix E) already demonstrates a clear reliability gap.
>
> That said, we agree that including [1] empirically would strengthen the work. If the paper is accepted, we plan to:
>
> 1. implement the invariant-feature baseline within our codebase, and
>
> 2. evaluate it under the same $T$, $\delta$, and attack settings, reporting both TPR and FPR for provenance detection.
>
> We will clarify this limitation and our planned future comparison in Section 5.1.
>
> ---
>
> **Weakness 2.** The attacks tested are mostly token-level and synthetic. There is no evaluation under semantic or paraphrasing-based attacks.
>
> **Response:**
> As noted in your comment, we have added paraphrasing experiments:
>
> - Section 5.3 and Table 2 evaluate robustness under T5-Paraphrase-Paws on both C4 and OpenGen across $\delta$ and $s_{\max}$.
>
> - Structured-Ours maintains FPR between 0.0 and 0.14 (often exactly 0), while RS-Watermark’s FPR remains in the 0.90–0.97 range.
>
> - TPR for Structured-Ours increases with $\delta$, indicating that stronger embedding improves resilience to semantic rewriting.
>
> These results directly address the concern about practical paraphrasing attacks.
>
> ---
>
> **Weakness 3.** The quadratic detection complexity may limit use in long-form or high-throughput contexts unless optimized or parallelized. Detection latency for different $s_{\max}$ is needed.
>
> **Response:**
> As described in our response to DpVy-Q3, Appendix C.3 provides a formal complexity analysis, and Appendix E.6 (Table 4) reports actual detection times. Even with longer codewords ($n = 63$), longer texts ($T = 500$), and $s_{\max} = 5$, detection completes within $\le 0.54$ seconds on our hardware. These results suggest that the method is usable for practical context lengths (hundreds of tokens) and moderate throughput.

---

> ### Author Response · Authors · 2025-11-21
> **Comment on Weakness 4.**
>
> **Weakness 4.**The paper would be stronger if it quantified how each component (block partitioning, window-shifting, designated verification) contributes to overall FPR reduction.
>
> **Response:**
> We added an ablation that isolates these components:
>
> 1. designated-only: no window shifting; accept only the designated codeword
>
> 2. shift-only: window shifting but accept any codeword
>
> 3. both: window shifting + designated-codeword verification (full method)
>
> Under 10% insertion attacks with $\delta = 6$ (figure/table included in the rebuttal), we observe:
>
> 1. designated-only: very low FPR ($\approx 0.005$–$0.015$) but moderate TPR ($\approx 0.36$–$0.43$), since desynchronization is not corrected
>
> 2. shift-only: TPR increases substantially ($\approx 0.73$–$0.75$) but FPR also rises ($\approx 0.14$–$0.13$), consistent with larger acceptance regions
>
> 3. both: TPR similar to or higher than shift-only, while FPR remains close to designated-only
>
> | model           | s_max | T200 TPR | T200 FPR | T200 Precision | T200 F1 | T500 TPR | T500 FPR | T500 Precision | T500 F1 |
> |-----------------|-------|----------|----------|----------------|---------|----------|----------|----------------|---------|
> | designated-only |   –   | 0.355    | 0.005    | 0.9861         | 0.5221  | 0.430    | 0.015    | 0.9663         | 0.5952  |
> | shift-only      |   1   | 0.205    | 0.030    | 0.8723         | 0.332   | 0.330    | 0.040    | 0.8919         | 0.4818  |
> | shift-only      |   3   | 0.540    | 0.035    | 0.9391         | 0.6857  | 0.745    | 0.130    | 0.8514         | 0.7947  |
> | shift-only      |   5   | 0.725    | 0.140    | 0.8382         | 0.7775  | 0.880    | 0.190    | 0.8224         | 0.8502  |
> | both            |   1   | 0.500    | 0.035    | 0.9346         | 0.6515  | 0.630    | 0.075    | 0.8936         | 0.739   |
> | both            |   3   | 0.730    | 0.050    | 0.9346         | 0.8202  | 0.815    | 0.105    | 0.8859         | 0.849   |
> | both            |   5   | 0.815    | 0.110    | 0.8810         | 0.8468  | 0.960    | 0.265    | 0.7837         | 0.8629  |
>
> This ablation confirms that designated-codeword verification is the primary driver of FPR suppression, while window shifting is responsible for recovering TPR under insertion/deletion. Block-wise embedding enables these mechanisms to operate independently per block.

---

### Official Review · Reviewer_DpVy · 2025-11-09

**Soundness:** 2
**Presentation:** 2
**Contribution:** 2
**Rating:** 2
**Confidence:** 4

**Summary:**

This paper proposes a multi-bit watermarking framework that aims to address the claimed high false positive rate (FPR) problem in existing methods. The authors introduce a block-wise error correction approach with window-shifting detection. While the motivation is reasonable, the paper contains fundamental misunderstandings of prior work and lacks rigorous experimental validation.

**Strengths:**

** Window-shifting detection for insertion/deletion robustness**

The window-shifting detection mechanism is a reasonable engineering solution to handle insertion/deletion-induced misalignment. While the core idea of scanning for valid patterns is not entirely novel, the systematic application of multi-bit watermarking with explicit synchronization recovery represents a practical contribution.

**Weaknesses:**

**1. Fundamental Misunderstanding of Prior Methods' Detection Protocol**

The authors claim that prior methods like Qu et al. (2025) and MPAC suffer from "unacceptably high false positive rates" (lines 54-57, 142-144). However, this criticism stems from a misunderstanding of how COUNT matrix-based multi-bit watermarking actually works.

As Qu et al. explicitly state in their Introduction: *"Given a text that is suspected to be LLM-generated, the watermark extraction function can be used to identify whether the text is watermarked. Furthermore, if the watermark is identified, the user ID can be extracted from the watermark."*

Similarly, Yoo et al. (MPAC) describe in Section 3.3: *"To distinguish between a watermarked text and a non-watermarked (human-written) text, we count the number of tokens assigned to the predicted message..."*

**The standard workflow is**:

   1. First, use the COUNT matrix to perform zero-bit detection (is watermark present?)
   2. Only if watermark is detected, then extract the multi-bit message

The authors' experimental comparison appears to skip the essential first step, directly applying message extraction to unwatermarked text. This fundamentally misrepresents the FPR of prior methods. A fair comparison must include the initial watermark presence detection step that these methods explicitly specify.

**2. Overstated Novelty Claims**

The paper claims several innovations, but careful examination reveals significant overlap with existing work:

- **"Code-agnostic design"** (lines 90): Yoo et al., Qu et al., and Feng et al. are all code-agnostic and compatible with different ECC schemes. This is not novel.
- **"Distributed codeword architecture"** (line 159): Yoo et al., Qu et al., and Feng et al. use hash functions to distribute multi-bit messages across different tokens/segments. The token-to-segment and bit-to-G/R list mapping described here is essentially the same mechanism.
- **"Incremental detection"** (lines 81-82): First, the idea of "graduated confidence assessment" and using multiple codeword segments already exists in prior methods employing COUNT matrices (Yoo et al., Qu et al., Feng et al.). Second, the claim "enabling graduated confidence assessment rather than binary detection" is confusing, as multi-bit watermarking is inherently non-binary by definition.
- **"Codeword assignment with key-derived mask"** (line 224): This XOR approach appeared in prior works (Fairoze et al, Feng et al.) but lacks proper citations.

**3. Insufficient Analysis of Core Design Mechanisms and Their Trade-offs**

The authors claim their method "achieves substantially improved TPR–FPR through incremental evidence collection with explicit insertion/deletion handling" (lines 159-193). However, the paper fails to establish a clear causal chain linking individual design components to the claimed improvements:

**Potential Contradictory Effects of Core Mechanisms**:
- **Window-shifting detection**: Intuitively, this technique should increase FPR by creating more alignment opportunities for spurious matches in non-watermarked text, even as it improves TPR by recovering misaligned codewords after insertions/deletions.
- **Designated codeword verification**: only accepts codewords matching the initially embedded
message, which should decrease both TPR and FPR.

While designated codeword verification appears to be the core innovation, the paper lacks critical analysis of how these opposing mechanisms interact. The two techniques have contradictory effects on both metrics, yet the paper claims simultaneous improvements in TPR and dramatic FPR reduction without providing:

   1. Trade-off analysis: How do these two mechanisms balance against each other? What are the optimal parameters for this balance?

   2. Ablation studies: What is the individual contribution of each component to the final performance?

   3. Computational efficiency analysis: Window-shifting requires trying multiple alignment positions. What is the computational overhead compared to prior methods?

Without such analysis, it remains unclear whether the improvements stem from the specific design innovations, parameter tuning, or other confounding factors in the experimental setup.

**4. Inconsistency in Evaluation Metrics: Mixing Detection Paradigms**
The paper employs a problematic double standard in its evaluation framework that undermines the validity of its comparative results:

- **For False Positive Rate** (Lines 357-361):
The paper evaluates prior methods using a binary detection criterion: any successfully decoded codeword triggers a positive detection. This is essentially 0-bit watermarking evaluation.

- **For False Negative Rate** (Lines 362-365):
The paper switches to a multi-bit perfect matching criterion: only exact recovery of the originally embedded message counts as successful detection. This is multi-bit extraction accuracy evaluation.

[1] Yoo, KiYoon, Wonhyuk Ahn, and Nojun Kwak. "Advancing beyond identification: Multi-bit watermark for large language models." Proceedings of the 2024 Conference of the North American Chapter of the Association for Computational Linguistics: Human Language Technologies (Volume 1: Long Papers). 2024.
[2] Qu, Wenjie, et al. "Provably robust multi-bit watermarking for {AI-generated} text." 34th USENIX Security Symposium (USENIX Security 25). 2025.
[3] Feng, Xiaoyan, et al. "BiMark: Unbiased Multilayer Watermarking for Large Language Models." Forty-second International Conference on Machine Learning.
[4] Fairoze, Jaiden, et al. "Publicly-Detectable Watermarking for Language Models." IACR Communications in Cryptology 1.4 (2025).

**Questions:**

1. Could the authors provide experimental results where prior methods (Yoo et al, Qu et al.) are evaluated with their intended detection protocol (i.e. **watermark existence detection based on total green tokens derived from COUNT matrix before message extraction**)?
2. Regarding "designated codeword verification.": **Does this mean that the embedded message is known during the verification process?** If known, why is verification necessary? If unknown, does it require iterating through the entire message set?
2. What is the **computational cost** of window-shifting detection, and how does it affect the false positive rate in practice?

---

> ### Author Response · Authors · 2025-11-21
> **Comment on Question 1**
>
> 1) RS-Watermark (Qu et al.)
>
> Our main comparison with RS-Watermark uses the detection protocol implemented in the authors’ official code, following the procedure described in Qu et al.:
>
> compute the COUNT matrix / green-token statistics for the segments
>
> apply the existence test (their threshold on the aggregated score)
>
> attempt RS decoding only if the existence test passes
>
> In our reproduction, we do not modify their thresholds or scoring rule. We only vary the corruption level (e.g., 10% synonym substitution) and the text length ($T \in {200,500}$). Under this protocol, the FPR remains very high. For example, on 200-token C4 text with 10% synonym substitution, RS-Watermark achieves TPR $\approx 0.97$ but FPR $\approx 0.925$, matching the results reported in the introduction and Section 5.2. These FPR values are measured at the existence-test output (before RS payload decoding), so they directly reflect the behavior of the intended detector.
>
> We will clarify this explicitly in the camera-ready version by adding:
> “RS-Watermark is evaluated using the authors’ official COUNT-based existence test with default thresholds, and FPR is measured at this binary decision stage.”
>
> 2) Yoo et al.
>
> Yoo et al. use invariant features (keywords and syntactic patterns) rather than an ECC-coded, per-token partition. Their detector is tailored to those features and reports a match rate of 49.2% for 32-bit payloads on long messages.
>
> A faithful, apples-to-apples reproduction in our setting would require:
>
> re-implementing their feature extractors for our models and datasets
>
> making several design choices that are not fully specified in their paper
>
> We were not able to find an official, end-to-end implementation aligned with our evaluation protocol. To avoid a potentially unfair or partially assumed re-implementation during the short rebuttal period, we treat Yoo et al. as a qualitative baseline and discuss it in Section 2.2 as a non-ECC multi-bit method with limited extraction accuracy on long sequences.
>
> If the paper is accepted, we plan to implement Yoo et al. in our codebase and add a dedicated appendix comparing its TPR/FPR trade-offs under our attack suite.

---

> ### Author Response · Authors · 2025-11-21
> **Comment on Question 2**
>
> Our framework supports both payload-carrying and zero-payload (pure provenance) modes.
>
> 1. Multi-bit payload setting.
> In this case, the verifier typically knows or can reconstruct the embedded message $m$, e.g., from server logs (session ID, user ID, timestamp, etc.).
> Given the secret key $K$ and $m$, the verifier deterministically recomputes the per-block designated codewords $E(m \oplus r(j))$,
> and then checks whether the decoded $\hat{c}^{(j)}$ equals this designated codeword for each block (Algorithm 2).
>
> 2. Zero-payload / provenance-only setting.
> We explicitly allow setting $m = 0^{k}$ and using the masks $r(j)$ as block-specific pseudo-messages; the detector remains unchanged (as noted in Section 3.1.1).
> In this case, the “message’’ is effectively part of the key.
>
> In both cases:
>
> 1. The detector never iterates over the message space. It checks only a single designated codeword per block, derived from $(K, m)$ (or from $K$ alone in the zero-payload mode).
>
> 2. Verification is still required because the detector only receives an arbitrary candidate text and must determine whether that text is compatible with the specific designated codeword sequence corresponding to this generation.
> This is exactly the provenance question: “Was this text produced by our system with this key / metadata?”

---

> ### Author Response · Authors · 2025-11-21
> **Comment on Question 3**
>
> **Complexity.**
> Appendix C.3 shows that detection complexity is
> $O(T \cdot s_{\max} \cdot n^{2})$,
> where $T$ is text length, $n$ is the code length, and $s_{\max}$ is the maximum circular shift.
> Bit extraction is $O(T)$; decoding is $O(n^{3})$ per block with standard algorithms, and the shift search adds a factor $s_{\max}$.
>
> Appendix E.6 (Table 4) reports wall-clock inference times from our implementation:
>
> For $T = 200$, $n \in {15, 31, 63}$, and $s_{\max} = 5$, detection takes 0.12–0.20 s.
>
> For $T = 500$, even with $n = 63$ and $s_{\max} = 5$, detection remains around 0.53 s.
>
> Thus, the method is fast enough for offline provenance checks and moderate-throughput settings.
>
> **Effect on FPR.**
> Theoretically:
>
> - Theorem 2 in Appendix D shows that window-shifting multiplies the single-shift false-positive probability $p_{0}$ by at most a factor
> $S = 2 s_{\max} + 1$ (union bound).
>
> - Theorem 3 shows that aggregating over many blocks with a match-ratio threshold $\theta$ keeps the overall FPR exponentially small in the number of blocks $M$.
>
> Empirically, we observe exactly this behavior:
>
> - As shown in Figures 3, 9, and in the tables in Appendix E.8, increasing $s_{\max}$ from 0 to 5 substantially improves TPR under deletion/insertion-like attacks, while FPR for Structured-Ours remains near zero (typically $\le 0.02$) across all $\delta$, models, and datasets.
>
> - By contrast, RS-Watermark’s FPR is dominated by its “any-codeword” decoding and remains around $0.9$ regardless of the shift budget.
>
> We also provide an ablation comparing:
>
> 1. designated-only (ECC, no shifts)
> 2. shift-only (shifts, but accept any codeword)
> 3. both (our full method)
>
> | model           | s_max | T200 TPR | T200 FPR | T200 Precision | T200 F1 | T500 TPR | T500 FPR | T500 Precision | T500 F1 |
> |-----------------|-------|----------|----------|----------------|---------|----------|----------|----------------|---------|
> | designated-only |   –   | 0.355    | 0.005    | 0.9861         | 0.5221  | 0.430    | 0.015    | 0.9663         | 0.5952  |
> | shift-only      |   1   | 0.205    | 0.030    | 0.8723         | 0.332   | 0.330    | 0.040    | 0.8919         | 0.4818  |
> | shift-only      |   3   | 0.540    | 0.035    | 0.9391         | 0.6857  | 0.745    | 0.130    | 0.8514         | 0.7947  |
> | shift-only      |   5   | 0.725    | 0.140    | 0.8382         | 0.7775  | 0.880    | 0.190    | 0.8224         | 0.8502  |
> | both            |   1   | 0.500    | 0.035    | 0.9346         | 0.6515  | 0.630    | 0.075    | 0.8936         | 0.739   |
> | both            |   3   | 0.730    | 0.050    | 0.9346         | 0.8202  | 0.815    | 0.105    | 0.8859         | 0.849   |
> | both            |   5   | 0.815    | 0.110    | 0.8810         | 0.8468  | 0.960    | 0.265    | 0.7837         | 0.8629  |
>
> Designated-only yields the smallest FPR but limited TPR;
> shift-only increases both TPR and FPR;
> the combined method achieves high TPR while keeping FPR close to designated-only.
>
> This confirms that window-shifting primarily improves TPR, and the designated-codeword test prevents FPR from growing significantly.

---

### Author Response · Authors · 2025-11-21
**Author Response**

We thank all reviewers for their careful reading and constructive feedback.
In response, we have revised the manuscript and added several new experiments and analyses:

**Paraphrasing / semantic attacks** using T5-Paraphrase-Paws (Section 5.3, Table 2). Our method (“Structured-Ours”) keeps FPR near zero across all $\delta$ and models, while RS-Watermark maintains FPR $\approx 0.9$ even though TPR is high.

**20\% mixed synonym attacks** combining insertion-like, deletion-like, and replacement-like edits (Appendix E.9, Tables 21–22). Structured-Ours retains low FPR, whereas RS-Watermark’s FPR remains near 1.0.

**Runtime / complexity analysis** of window-shift detection (Appendix C.3 and E.6, Table 4), showing detection times well below one second even for $T=500$, $n=63$, $s_{\max}=5$.

**Codeword-length ablation** across $n={15,31,63}$ (Appendix E.7, Figure 14 and Tables 5–8), clarifying the TPR–FPR trade-off and supporting our default choice of $n=31$.

**Threshold ablation** on the number of required matched codewords (Appendix E.5, Figures 12–13), showing how increasing the threshold trades TPR for even smaller FPR.

**Text-quality evaluation** (Appendix E.3, Table 3) with PPL, BLEU, and BERTScore, demonstrating that our soft variant preserves text quality better than RS-Watermark at all $\delta$.

**LLM usage statement** added in Appendix A as requested.

We also clarified the role of designated-codeword verification, the independence assumption in the theoretical analysis, and the practical detection scenario (Section 3.1 and Appendix D). Below we provide point-by-point responses.

---

### Note · Program_Chairs · 2026-01-17
**Submission Desk Rejected by Program Chairs**

The following references in this submission do not refer to real documents and/or have major errors in bibliographic information:

 Alexander Pan, June Tong Shern, Alex Tamkin, Jasmine Song, Yuntao Zhang, Sarah Saleh, Deep Ganguli, and Aidan Clark. Risk sources and risk management for generative ai. arXiv preprint arXiv:2310.07782, 2023.